



# HRRTLE (High Resolution Runoff and Transmission Loss Estimator): a novel tool for mapping connectivity of runoff in ephemeral stream networks to aid the siting of water harvesting structures

Robert G. Delaney[1], George A. Blackburn[1*1], Andrew M. Folkard[1], James D. Whyatt[1]

[1]Lancaster Environment Centre, Lancaster University, Lancaster, LA1 4YQ, UK

*Correspondence to*: Robert G. Delaney (rdelaney@lancaster.ac.uk)

**Abstract.** Water harvesting is predominantly carried out in arid and semi-arid regions. Site selection studies often rely on a methodology that calculates runoff using curve numbers to generate runoff maps. These maps, typically used as part of a multi-criteria selection process, identify areas conducive to the siting of water harvesting structures. However, traditional runoff maps do not account for transmission losses that occur along the surface flow path to the catchment outlet, and these losses can be significant in arid and semi-arid regions. Here we introduce a methodology that incorporates a curve number runoff method while also addressing transmission losses.

Our approach, utilising three global datasets, was validated against observed runoff data from 28 catchments worldwide, and infers hydraulic characteristics of both overland and channel flow from curve number values. This involves leveraging the curve number dataset twice: initially for calculating runoff and subsequently for forecasting transmission losses. The outcomes include a runoff connectivity map, at a spatial resolution of 250 m × 250 m, presenting the runoff depth (in mm) for each pixel based on the direct runoff generated at that pixel and reaching the catchment outlet. This connectivity map aids planners in comprehending the dynamics of surface runoff towards a catchment outlet, assisting in identifying potential locations for future water harvesting structures.

The process integrates 38 years of precipitation data, enabling predictions not only for average annual runoff but also for the return periods of various annual runoff volumes. Despite the simplicity of the model, a positive Nash-Sutcliffe efficiency value was observed in 11 out of the 28 catchments.

---

\* deceased





# 1 Introduction

Water harvesting – the collection of runoff water for productive purposes (Critchley and Siegert, 1991) – is a common practice in arid and semi-arid regions. Beyond their primary role of collecting and storing water for domestic, agricultural or industrial use, water harvesting structures can have many other benefits, including increased plant biomass production,

recharge of aquifers, reduced soil erosion, and flood mitigation (Gupta, 1994; Abdeldayem et al., 2020; Parimalarenganayaki, 2021; Strohmeier et al., 2021). There are many factors that need to be considered when deciding whether a location is appropriate for development as a water harvesting site, and several different methodologies have been deployed to identify sites. These include methods applicable to different types of water harvesting systems, including check dams (Patel et al., 2015, Ettazarini, 2021), structures located in gullies (Li et al., 2020), and small dams located on the

surface or underground (Forzieri et al., 2008). Surprisingly, the prediction of runoff (i.e., water volume inflow to a water harvesting storage site from its catchment) is not routinely used in site selection. Adham et al. (2016a) reviewed water harvesting site selection studies and found only 13 of 48 included runoff as a site suitability criterion, while the most frequently used biophysical criterion was slope (40 of 48 studies). Quantifying runoff volume to a potential site is crucial to determine whether a scheme will receive sufficient water on an annual basis to fulfil its intended purpose; help determine the

height of the storage structure that needs to be constructed (Stephens, 2010); and ascertain if excessive inflows will be problematic. Locations where the ratio of total volume of inflow to storage capacity is close to one are optimal for siting water harvesting structures (Adham et al., 2016). Hence, an important principle on which the work reported here is based is that the ratio of mean annual inflow volume to water harvesting storage volume is a key design metric and should be incorporated as one of the most important biophysical criteria in any water harvesting siting methodology.


Predicting the harvested water volume at a potential site requires knowledge of its catchment's area, rainfall, and rainfall-runoff relationship (Critchley and Siegert, 1991). Because catchments differ in terms of size, topography, geology, and land cover, the rainfall-runoff relationship will vary between them. Rainfall patterns can also change significantly, even between nearby catchments. Moreover, climate change is affecting mean precipitation and evaporation with "seasonally variable

regimes becoming more variable" (Konapala et al., 2020, p.1). Therefore, prediction of harvested water volumes for proposed storage sites requires contemporary data specific to the catchment in question.

Several methods have been used to quantify runoff in water harvesting site selection studies. These include the empirical formula of Tixeront (Mechlia et al., 2009), the Finkel method (Elewa et al., 2012), the Watershed Modeling System

conceptual model (Jabr and El-Awar, 2004) and the Soil Conservation Service Curve Number (SCS-CN) methodology (Gupta et al., 1997; Senay and Verdin, 2004; Kadam et al., 2012; Mugo and Odera, 2019; Shalamzari et al., 2019). Their outputs typically comprise of runoff maps (Senay and Verdin, 2004), predictions of total catchment runoff (e.g., Gupta et al., 1997) and runoff coefficients (e.g., Ramakrishnan et al., 2009). Runoff maps allow the ratio of annual runoff to available





storage volume to be calculated for potential water harvesting sites (Sayl et al., 2019). Presented as a thematic layer, with
runoff classified ordinally (e.g., low, moderate, or high) or fully quantitatively (e.g., depth, annual flood volume) they have
been incorporated into GIS-based site selection methodologies in various ways. For example, De Winnaar et al. (2007) and
Nagarajan et al. (2015) created maps showing zones of low, moderate, and high runoff potential. Sayl et al. (2019) and Al-
Ghobari et al. (2020) created maps of annual flood volume and potential runoff depth. Haile and Suryabhagavan (2019)
incorporated a thematic map of runoff depth into a Fuzzy Logic model as part of a GIS-based approach for identifying
potential rainwater harvesting sites. To improve the effectiveness of simulating the final runoff map of the watershed Karimi
and Zeinivand (2021) used a distributed spatial-physical based model with 594 "subwatersheds" to create an annual runoff
depth map to locate potential rainwater harvesting sites whilst accounting for daily temperature and evapotranspiration.

The method used most commonly for runoff calculation in water harvesting site selection is the SCS-CN methodology,
which was first introduced in 1956 (Mishra et al., 2012). It can be described as a conceptual model supported by empirical
data, which is used to estimate the volume of direct runoff (i.e., runoff generated by rainfall, rather than from baseflow)
generated at locations within a catchment from rainfall depth, using an empirical parameter known as a "curve number"
(CN), values of which are determined based on soil type and soil cover (e.g., vegetation or crops, vegetative debris, built
environment surface materials) (Ponce and Hawkins, 1996). CN is essentially a measure of land surface permeability and
therefore of sub-surface potential moisture retention capacity, and by extension the potential for runoff to be generated by
precipitation. The SCS-CN methodology calculates runoff using a CN value by first finding the soil water retention capacity,
$S$, using:

$$S = 25.4 \left( \frac{1000}{CN} - 10 \right),$$   **(1)**

where $S$ is the maximum soil water retention (mm), and CN is the curve number (dimensionless). From this, runoff generated
is computed using:

$$Q = \frac{(P - 0.2S)^2}{(P + 0.8S)} \quad if \ P > 0.2S$$

$$Q = 0 \qquad if \ P \leq 0.2S \ ,$$   **(2)**

where $Q$ is the direct runoff (mm), and $P$ is the storm rainfall (mm). Curve number rainfall-runoff models are best used for
ungauged catchments when runoff is the only output needed (Sitterson et al., 2017). As water harvesting site selection
planners typically deal with ungauged catchments it is not surprising that so many water harvesting selection studies use the
SCS-CN methodology to compute runoff. While this method is appropriate for such purposes, difficulties do remain.
Notably, as it was formulated for use on small agricultural catchments (Soulis, 2021), its application to larger catchments
needs to take into account the tendency for runoff efficiency to decrease as catchment area increases (Karnieli et al., 1988).




To predict the water storage yield at a particular location from precipitation in its catchment, in addition to knowledge of catchment area, rainfall and rainfall-runoff relationships, understanding is also required of the transmission losses – to infiltration, evaporation or other processes – experienced by the runoff as it travels from its points of creation (where the precipitation falls) to the proposed storage location. These losses are typically high in arid regions, where water harvesting is commonly practised (McMahon and Nathan, 2021). Hughes and Sami (1992) estimated total transmission losses of 22 % and 75 % for two rainfall events in a semi-arid ephemeral channel system located in Africa with transmission losses largely taking the form of infiltration of water into the ground. For a semi-arid basin in Brazil, Toledo et al. (2020) stated that transmission losses accrued at a rate of 2.7 % for every kilometre of river system. Consequently, runoff maps which do not include an allowance for transmission losses cannot be verified against observed flow data, should they exist. Thus, another difficulty in using the SCS-CN method in larger catchments is obtaining data quantifying the land surface conditions over which transmission losses occur at sufficient spatial resolution. Typically, modelling a catchment to incorporate such transmission losses involves aggregating land into sub-catchments with uniform runoff-loss characteristics. This "lumped" approach reduces spatial variability. Remote sensing offers an ever-increasing availability of high spatial resolution data products that can address this problem. The method described in this paper aims to exploit this to create high-spatial resolution runoff maps whilst incorporating transmission losses.

The aim of this study is to develop and test a novel procedure to create maps showing the mean annual runoff from locations within arid and semi-arid catchments to collection points that takes into account transmission losses at high spatial resolution, i.e., at the pixel resolution of currently available remote sensing data, rather than at the much coarser sub-catchment scale that has typically been used to date. The intention is that this procedure can be used to aid the siting of water harvesting structures in regions where on-the-ground data is sparse. This aim is addressed through the following objectives:

1. Create a model to compute generated runoff using global precipitation and curve number datasets.
2. Model flowpaths from points where runoff is generated to the catchment outlet.
3. Develop a transmission loss model to determine the proportion of runoff reaching the outlet.
4. Evaluate model results against observed runoff data, including evaluating the effect of incorporating transmission losses by comparing results for model runs with and without them incorporated.
5. Examine the characteristics of catchments best modelled by the procedures developed.

The novel contributions of this work lie in the use of fully distributed data sets, rather than the lumped approach taken previously, and in the novel method put forward for calculation of transmission losses. The approach taken to calculating transmission losses is based on the following argument. In arid and semi-arid regions, there are far fewer rainy days than in humid regions. Only some rainy days create direct runoff. Even fewer rainy days are responsible for runoff reaching a collection point. So, in these regions, where water harvesting is largely practiced, there are only brief periods when runoff is





being generated and transferred to a candidate water harvesting site downstream. Within such ephemeral systems, baseflow is less significant, or largely absent, compared to more humid regions. The method described here exploits these characteristics of arid zone hydrology, generating runoff using daily precipitation data, while surface flow (and hence transmission loss) is modelled as a singular annual event. Such an approach negates the need to route hydrographs hence sub-basins do not have to be created and catchments can be modelled at relatively high spatial resolution.

The rainfall–runoff yield model effectively consists of two components. The first component generates direct runoff from daily precipitation data using the SCS-CN method with curve number values extracted from a global dataset (Jaafar et al., 2019). The second component calculates transmission losses over flowpaths from cells where the runoff is generated to the candidate water harvesting storage site, at high spatial resolution. Outputs include a runoff connectivity map (RCM) of annual runoff depth reaching the storage site, and the predicted mean annual runoff volume. These two components are combined in the "High Resolution Runoff and Transmission Loss Estimation" tool, or HRRTLE (pronounced "hurtle").

## 2 Materials and Methods

### 2.1 Method summary

A set of catchments in arid or semi-arid climate zones were identified for which the necessary data sets – elevation (as a digital elevation model, DEM), curve number, rainfall, and discharge – were available. The position of the gauging station used to gather discharge data determined the outlet of each catchment, hence catchments became proxies for candidate water harvesting catchments for model development purposes, with their outlets (gauging station location) acting as places for potential collection and storage sites for the harvested water. Each catchment in turn was represented as an array of 250 m × 250 m cells and characterised in terms of its size, shape, and elevation. Using a long-term precipitation dataset and a global curve number dataset, runoff generated directly by precipitation was calculated for all cells within each catchment at a daily resolution using the SCS-CN procedure. These were summed to give an annual value of runoff (in mm) generated at each cell. This was followed by the calculation of transmission losses – quantified as transferral ratios, the fraction of generated runoff reaching the catchment outlet on an annual time scale – which involved several stages. Firstly, analysis of pixel-scale flow accumulation, derived from the catchment's DEM, was used to define the catchment's stream network, and distinguish it from the rest of the catchment, where water fluxes were assumed to occur via overland flow. Flowpaths were then defined between each cell and the catchment outlet and classified into sections of in-stream and overland flow. Transferral ratios for the in-stream segment of each flow pathway were determined by considering the curve number and flow transit time. The transit time was derived from the length of the in-stream flow path and the velocity of the flow. The flow velocity was computed using Manning's equation, utilising proxies for hydraulic radii and roughness coefficients, which were determined based on available data and a set of assumptions and approximations. For the overland section of each flow path, transferral ratios were again calculated as a function of curve number and flow transit time, of the same form as that for in-stream flow,





but with different values of the curve number power law index and travel time constant. Overland flow transit time was

calculated from overland flow path length and flow speed, and flow speed was calculated as a function of curve number and

155 topographic slope, again based on the available data and a set of assumptions and approximations. The overall transferral

ratio was then calculated as the product of the in-stream and overland flow section transferral ratios. This was multiplied by

the annual runoff value for each cell, and the total modelled discharge at the catchment outlet calculated by:

$$Q_a = \left(\frac{\sum_{i=1,j=1}^{i=m,j=n}(Q_c(i,j)TR_c(i,j))}{1000N}\right)A_c \;, \tag{3}$$

where $Q_a$ is the annual discharge (m³ y⁻¹); $i$ and $j$ are cells in the X and Y directions respectively, $i$ running from 1 to $m$, and $j$

running from 1 to $n$; $Q_c(i,j)$ is the annual direct runoff (expressed in rainfall depth equivalent, mm y⁻¹) generated at cell $(i,j)$;

160 $TR_c(i,j)$ is the overall transferral ratio for cell $(i,j)$; $N$ is the total number of cells in the catchment; and $A_c$ is the total

catchment area (m²). A schematic diagram of the main stages of the HRRTLE tool is presented (**Figure 1**).

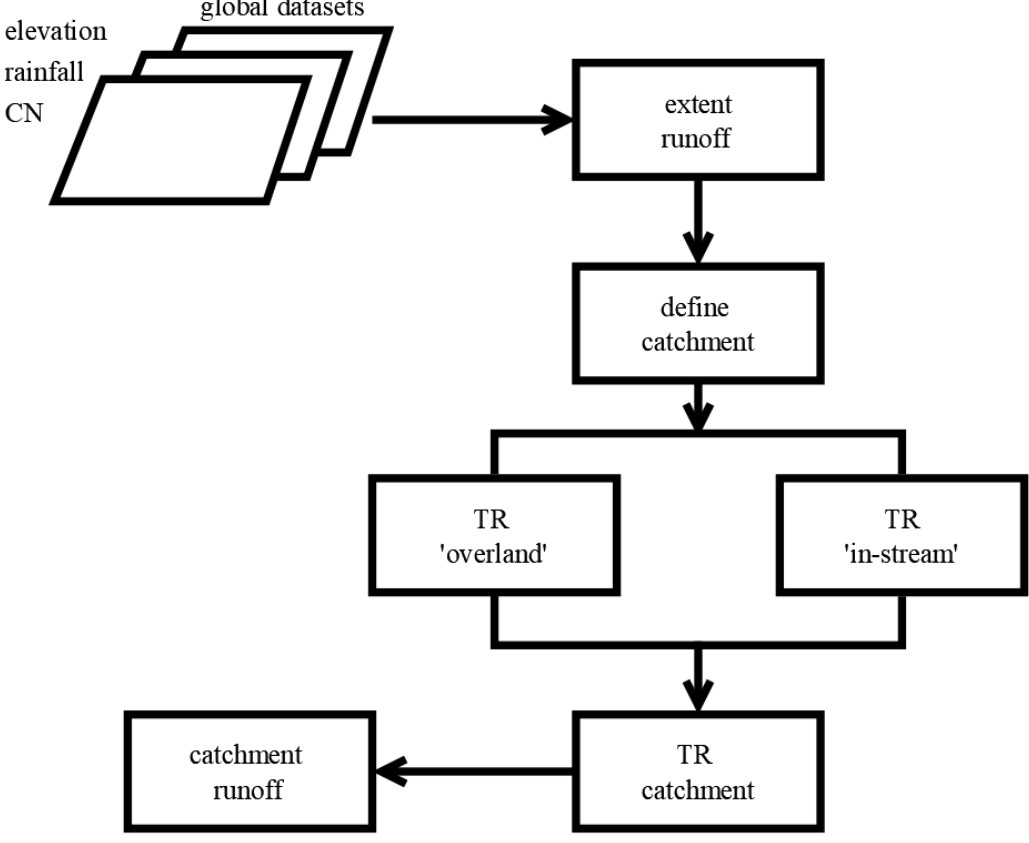

165 **Figure 1. Schematic of the main stages of the HRRTLE process [CN = Curve Number; TR = Transferral Ratio].**





The model was calibrated by comparing its outputs with the outflow data available for each catchment and adjusting the values of the power law index and time constant in the transferral ratio equations to optimise the model results' fit to the observed data. An assessment was then made of which types of catchments the model worked best for, why, and how the model might therefore be applied in practice, and improved in future research.

### 2.2 Methods in detail

### 2.2.1 Catchment area selection and characterisation

Observed catchment outflow records from the Global Runoff Data Center database (GRDC, 2020) were scrutinised to identify catchments located primarily in arid or semi-arid regions with records spanning four decades from the early 1980's and missing <3 % daily values. Once a catchment was identified as a potential study area, three catchment morphometric parameters were computed. A GIS was used to define the catchment area from a DEM (USGS EROS, 2018), with a spatial resolution of 3 arc-seconds for global coverage. The catchment form factor (CFF), defined as the ratio of catchment area to the square of the basin length (Patel et al., 2015), and the Height Above Nearest Drainage (HAND) were then computed from the DEM – see Sect. 2.2.3 below for details of how the drainage network was identified. Each catchment was classified based on its area, CFF and HAND (**Table A1**). A subset of 28 catchments (**Figure 2**) that offered a range of permutations of area (large, intermediate or small), shape (CFF), and elevation (HAND) were then selected for the purposes of model development (**Table A2**). Of these, 15 are in South Africa, 7 in Australia, 3 in the USA, 2 in Brazil and 1 in Israel.

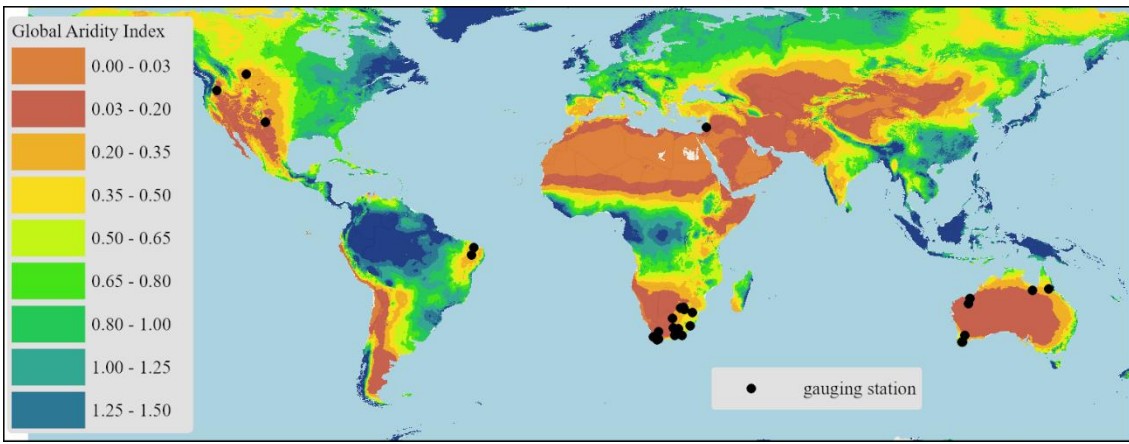

**Figure 2: Map showing locations of runoff gauging stations used. Each station represents the outlet of a modelled catchment. Basemap of Global Aridity Index (Zomer and Trabucco, 2022).**

To further characterise the selected catchments, additional datasets were obtained. The mean aridity index was found for each catchment using a global aridity dataset (Zomer and Trabucco, 2022). Mean rainfall was derived from a Global Precipitation Climatology Centre (GPCC) 1° resolution dataset (Ziese et al., 2022). The baseflow percentage (i.e., the





proportion of runoff at the catchment outlet that is derived from baseflow, rather than direct runoff generated by precipitation) was computed for each catchment by inputting the GRDC daily runoff data into a baseflow index calculator (U.S. Geological Survey, 2016). The runoff efficiency was calculated for each catchment by taking the ratio of the long-term runoff depth (mm) from the GRDC records to the mean depth (mm) obtained from the precipitation dataset. The mean peak runoff month was found by analysing observed runoff data. Land cover (spatial resolution ~1 km) for each catchment was obtained from global land cover datasets for the year 2000 (Eva et al., 2003; Latifovic et al., 2003; Mayaux et al., 2003; Mayaux and Bossard, 2003; Tateishi et al., 2003).

### 2.2.2 Direct runoff calculation

The spatial resolution of the GPCC precipitation data was enhanced from 1° (approximately 110 km on a great circle) to 250 m using linear interpolation, to create a precipitation dataset, $P$ (mm) that matched the spatial resolution of the CN dataset used (see below). Antecedent precipitation values, $P_{\Sigma 5}$ (mm), were then calculated for each day at each 250 m cell by summing the precipitation from the five preceding days. The antecedent moisture condition (AMC) was then assigned to each cell for each day based on the value of $P_{\Sigma 5}$ for that day and the season (**Table 1**), following Silveira et al. (2000).

**Table 1.** Selection of antecedent moisture condition (AMC) using antecedent precipitation ($P_{\Sigma 5}$) and season.

| AMC | dormant season $P_{\Sigma 5}$ (mm) | growing season $P_{\Sigma 5}$ (mm) |
|---|---|---|
| I (dry) | <13 | 36 |
| II (normal) | 13 to 28 | 36 to 53 |
| III (wet) | >28 | >53 |

Curve number values were taken from a global curve number dataset with a spatial resolution of 250 m (Jaafar et al., 2019), which provided CN values for the three AMC groups defined in **Table 1**. The appropriate CN value was selected according to the AMC value derived from **Table 1** and assigned to each cell in the extent. The daily direct runoff was then calculated from CN and $P$ for each cell using the SCS-CN equations Eq. (**1**) and Eq. (**2**).

### 2.2.3 Transferral ratio calculations

To determine catchment outflow from the direct runoff values for each cell in the catchment, knowledge of transmission losses is needed. These quantify how much of the direct runoff is lost – primarily to infiltration but also to evaporation and other processes (e.g., uptake by plants or animals) – on its journey from its source cell to the catchment outlet. Here, this is quantified in the form of transferral ratios – the fraction of the runoff that makes it to the catchment outlet. The transferral ratios are calculated by dividing the catchment into cells that form its drainage stream network (where fluxes are denoted "in-stream flow") and the rest of the catchment (where fluxes are denoted "overland flow"). For runoff from each cell, partial





transferral ratios are calculated for both that part of its journey that occurs as overland flow ($TR_o$), and that part that occurs as in-stream flow ($TR_n$). The overall transferral ratio ($TR_c$) is then calculated as the product of $TR_o$ and $TR_n$.

Stream network identification and parameterisation

To identify the cells that formed the catchment's stream network, the spatial resolution of the DEM was increased from 3 arc-seconds to 250 m by unifying and filling its tiles in a GIS environment (ArcGIS Pro 2.8) effectively matching its spatial resolution to that of the curve number and precipitation data. D8 flow direction value for each cell was then derived by identifying the neighbouring cell whose elevation was lowest. The D8 flow direction values were then used to calculate an
unweighted flow accumulation value for each cell, defined as the total number of cells flowing into it.

Having calculated the flow accumulation value for each cell, the catchment's stream network was defined as being made up of all cells with a flow accumulation value greater than 65. This was based on the assumption that the threshold drainage area required to initiate stream formation and maintenance in arid or semi-arid zones is 4.05 km$^2$ (Gao et al., 2019), which
approximates to 65 cells of size 250 m × 250 m, to the nearest whole number. Rasters of the following variables covering all cells in the stream network were then created within the GIS: the height above the catchment outlet, $HACO_n$ (m); the horizontal distance to the catchment outlet along the stream network, $HFD_n$ (m); the mean downstream slope of stream network cells, $S_n$ (m m$^{-1}$), found by dividing $HACO_n$ by $HFD_n$, and the stream network mean downstream length, $L_n$ (m), calculated by taking the square root of the sum of the squares of $HACO_n$ and $HFD_n$.


In-stream flow transferral ratio calculation

The transferral ratio for that part of a cell's runoff's journey to the catchment outlet that travels down the stream network, $TR_n$, was calculated as:

$$TR_n = \left(\frac{CN_{fd,n}}{100}\right)^{p_n} e^{k_n T_n} \qquad (4)$$

Of the terms on the right hand side, $CN_{fd,n}$ – the stream network mean annual curve number value for flood-days
(dimensionless) – and $T_n$ – the stream network travel time (days) are calculated, whereas $p_n$ – the "curve number power law index" (dimensionless) – and $k_n$ – the "travel time constant" (dimensionless) – are parameters whose values are adjusted in the calibration process whereby the calculated outflow is compared with observed outflow data for each catchment. Thus, Eq. (4) encapsulates the assumptions that more runoff will reach the outlet as curve number increases (which follows from the definition of curve number), and the longer the runoff takes to reach the outlet, the more of it will be lost along the way.
Mathematically, these effects have been assumed to follow power law and exponential relationships, respectively.

Of the two calculated terms on the right hand side, the stream network mean annual curve number value for flood-days ($CN_{fd,n}$) was calculated as the mean CN value for stream network cells in the runoff's path for all days in the year when direct





runoff occurred (i.e., when $P_{\Sigma5} > 0.2S$). The calculation of the stream network travel time is more complex and is described
in the following section.

Stream network travel time calculation

The stream network travel time to the catchment outlet was calculated using:

$$T_n = L_n/(86400V_n) \,, \tag{5}$$

where $T_n$ is the network mean downstream transit time (days), $L_n$ is the stream network mean downstream length (m), and $V_n$
is the stream network mean downstream velocity (m s$^{-1}$). Calculation of $L_n$ is described above. Calculation of $V_n$ is based on
Mannings equation for open channel flow (e.g., Chow et al., 1988) shown by:

$$V_n = \frac{1}{n}R^{2/3}S_f^{1/2} \tag{6}$$

where $n$ is Manning's roughness coefficient (m$^{-1/3}$s), $R$ is the hydraulic radius (m), and $S_f$ is the friction slope
(dimensionless). Here, $S_f$ is assumed to be equal to the mean downstream slope of stream network cells, $S_n$ (see above), and
the – more complex – procedures for calculating $n$ and $R$ are as follows.


To calculate values for $n$, it is first noted that there is a relationship between land use, CN and Manning's $n$. For example,
natural forests have relatively low CN values and relatively high Manning's $n$ values, whereas bare land has relatively high
CN values and relatively low Manning's $n$ values (Schwab et al., 1981 cited in Tarigan, 2016). Using this principle, that
higher CN values are associated with lower hydraulic resistance to surface flow, a proxy for Manning's $n$ was established by
creating a linear relationship between it and CN. This was done by noting that the maximal Manning's $n$ value is
approximately 0.15, which is associated with "very weedy reaches, deep pools, or floodways with heavy stand of timber and
underbrush", while its minimal value is approximately 0.025, associated with "clean, straight, full stage, no rifts or deep
pools" (Das, 2016). Similarly, CN values vary from a minimum of approximately 20 to a maximum of 100. By associating
the maximal Manning's n with the minimal CN, and vice versa, and applying a linear regression between these two points,
the relationship

$$n = -0.0018CN + 0.185 \tag{7}$$

was derived. Using the stream network mean annual curve number value for flood-days ($CN_{fd,n}$), values of Manning's n can
thus be estimated and used in Eq. (6) to calculate flow speed and thus stream network travel time.

The final element of Manning's equation, the hydraulic radius, R, is defined as A/P, the ratio of the stream flow cross-
sectional area, A, to its wetted perimeter, P. These stream parameters cannot be determined directly from a DEM, or from
any of the other input data sets that are commonly available. The parameter most closely related to A and P that may be
derived from the available data is stream channel width, W. The precise relationship between W and A and P will vary, but





for small streams in arid or semi-arid zones, we assumed that the channels would be approximately triangular and have depth W/12. The decision to select a depth of W/12 was influenced by the common use of a width-to-depth ratio of 12 when

delineating natural rivers (Rosgen, 1994). This choice was anticipated to accommodate a diverse range of river types. Thus, we estimated the hydraulic radius using:

$$R = \frac{\frac{W}{2} \times \frac{W}{12}}{2\left(\left(\frac{W}{2}\right)^2 + \left(\frac{W}{12}\right)^2\right)^{0.5}} = \frac{W}{2\sqrt{148}} \approx \frac{W}{24} \tag{8}$$

We derived channel widths by assuming that they were linearly related to the flow accumulation value for each stream network pixel, the calculation of which is described in this section, above. Thus:

$$W_n = aFA_n + b \tag{9}$$

where $a$ and $b$ are empirical constants that require determination. This was carried out by manually measuring 20 stream

widths using an ArcGIS Pro World Imagery basemap (Source: Esri, Maxar, Earthstar Geographic, and the GIS User Community) from locations within each catchment selected to cover a range of flow accumulation values. These widths were regressed against their locations' flow accumulation values, to give catchment specific values for $a$ and $b$. A list of catchment codes alongside the values of these constants is provided in the **Supplementary Materials**.

Overland flow transferral ratio calculation

All catchment cells not designated as stream network cells, i.e., all cells with a flow accumulation value of 65 or less, were classified as "overland flow" cells. The calculation of the transferral ratio for runoff flowing through these cells followed the same general approach as that laid out above for in-stream cells, but required some different assumptions and approximations to be made to reflect the different nature of the conditions. Using the same assumptions as those represented

in Eq. (**4**) for stream network cells, the overland transferral ratio, $TR_o$, was calculated as:

$$TR_o = \left(\frac{CN_{fd,o}}{100}\right)^{p_o} e^{k_o T_o} \tag{10}$$

where $CN_{fd,o}$ is the mean CN value along the overland flow path for flood days; $p_o$ is the overland curve number power law index; $k_o$ is the overland travel time constant; and $T_o$ is the overland travel time (days). For each overland cell, the travel time for runoff to reach the catchment outlet, $T_o$, was calculated, analogously to Eq. (**5**), as:

$$T_o = L_o/(86400V_o) \tag{11}$$

where $L_o$ is the mean downstream overland length (m), and $V_o$ is the mean downstream overland velocity (m s$^{-1}$). $L_o$ was

calculated as the square root of the sum of the squares of HAND$_o$ and HFD$_o$, whose definition and derivation are described previously in this section. $V_o$ was calculated using a method based on the shallow concentrated flow equations. For unpaved





(grassed waterway) and paved surfaces, respectively, these give the relationships between flow speed, $V$, and along-flow slope, $S_f$, as (Cronshey et al., 1985) by:

$$V = 4.9178 S_f^{0.5} \tag{12}$$

and

$$V = 6.1960 S_f^{0.5} \tag{13}$$

By adopting a similar principle to that used for relating CN to Manning's coefficient in Eq. (**7**) above, a linear relationship was assumed between $CN_{fd,o}$ and the coefficients in Eq. (**12**) and Eq. (**13**). The coefficient for the rough, unpaved surface, 4.9178 in Eq. (**12**), was given an equivalent $CN_{fd,o}$ value of 60, while that for the smoother, paved surface, 6.1960 in Eq. (**13**), was given an equivalent $CN_{fd,o}$ value of 100. This recasts Eq. (**12**) and Eq. (**13**) as:

$$V_o = \left(0.0325 CN_{fd,o} + 2.95\right) S_o^{0.5} \tag{14}$$

where $V_o$ is the mean downstream overland velocity (m s$^{-1}$), $CN_{fd,o}$ is the overland mean downstream annual curve number
for flood-days (dimensionless), and $S_o$ is the mean downstream overland slope (m m$^{-1}$).

Overall transferral ratio

For each cell within the catchment, the overall transferral ratio, i.e., the proportion of rainfall running off from that cell that reached the catchment outlet, was calculated as:

$$TR_c = TR_n \times TR_o \tag{15}$$

where $TR_c$ is the catchment transferral ratio; $TR_n$ is the in-stream transferral ratio; and $TR_o$ is the overland transferral ratio (which is defined as 1 if the cell in question is part of the stream network). This was then multiplied by the (annual) direct runoff for the cell to give the model's estimate of that cell's contribution to the annual runoff at the catchment outlet (i.e., the harvested water if the catchment outlet represents a potential water harvesting location).

**2.3. Model performance**

The performance of the HRRTLE tool was evaluated used two commonly used measures, the Nash-Sutcliffe simulation efficiency (NSE) and percentage bias (Pbias), which were calculated, respectively, as:

$$\text{NSE} = 1 - \frac{\sum_{i=1}^{n}\left(Q_{i,obs} - Q_{i,cal}\right)^2}{\sum_{i=1}^{n}\left(Q_{i,obs} - \overline{Q_{obs}}\right)^2} \tag{16}$$

and





$$ \text{Pbias} = \left[ \frac{\sum_{i=1}^{n}(Q_{i,cal} - Q_{i,obs})}{\sum_{i=1}^{n}(Q_{i,obs})} \right] \times 100 \text{ \%} \tag{17} $$

where $n$ is the total number of events, $Q_{i,obs}$ is the observed flow, $Q_{i,cal}$ is the calculated flow, both at time $i$, and $\overline{Q_{obs}}$ is the
average observed flow (Cirilo et al., 2020). NSE is commonly used to assess the predictive abilities of hydrological models,
and generates values that can range from -∞ to 1. An NSE value of 1 indicates a perfect model, with zero mean difference
between observed and calculated flows. NSE = 0 implies that the model has no greater predictive power than simply
assuming constant flow equal to the observed mean. NSE < 0 means that the model has worse predictive power than the
observed mean flow (Knoben et al., 2019). Pbias measures the tendency of the calculated flows to be larger or smaller than
their observed counterpart flows (Yapo et al., 1996), with Pbias > 0 indicating that the model underestimates the observed
flows, and vice versa (Mendoza et al., 2021). While achieving a bias of zero would be an ideal target for a model for the sake
of scoping potential water harvesting (especially for ungauged catchments), Moriasi et al. (2007) and Abbaspour et al.
(2015) suggest that an absolute value of Pbias less than 25 % indicates good model performance. Here, as we are working
with ungauged catchments, we argue that this criterion for good performance should be somewhat relaxed, and thus take an
absolute value of Pbias less than 50 % to indicate a threshold between adequate and inadequate performance.

HRRTLE was applied to each of the 28 catchments and the calculated flows evaluated against observed runoff records. In
each case, in order to assess the value of applying the transferral ratio calculations described above, the model was run both
with and without transmission losses, $TR_c$. If the values of NSE and Pbias improved when $TR_c$ was applied, this would imply
that its use was beneficial for model accuracy. Four parameters ($k_n$, $p_n$, $k_o$, $p_o$) from the stream network and overland
transferral ratio equations, Eq. (4) and Eq. (10), were adjusted during the calibration stage. Each time one of these
parameters was adjusted, the model was re-run and the NSE recorded. Once the NSE could not be increased by adjusting a
particular parameter, the next parameter was used. Once a combination of parameter values whose NSE value could not be
improved had been found via this process, it was recorded and subsequently used in the model validation stage. Observed
runoff data from even years was used in the calibration stage, while odd years' data was used for validation. The number of
years of data used for the calibration and validation stages, together with the optimum transferral ratio parameter values used
in the validation stage for each catchment are presented in **Supplementary Materials**.

## 3 Results

### 3.1. Model output

Runoff connectivity maps (RCMs) were generated by the HRRTLE modelling process to visually communicate the
distribution of runoff contributions to the catchment outlets. An example of these maps is presented in **Figure 3** for





catchment AUNP, which has its outlet at a gauging station on the Shaw River in northern Western Australia. In this case, 18 years of validation model outputs were used to create the RCM, which shows the annual runoff depth (mm) for each cell that

reaches the catchment outlet, taking transmission losses into account. The spatial resolution of the map is 250 m × 250 m. The map shows the importance of proximity to both the outlet and the stream network for maximising outlet runoff contributions.

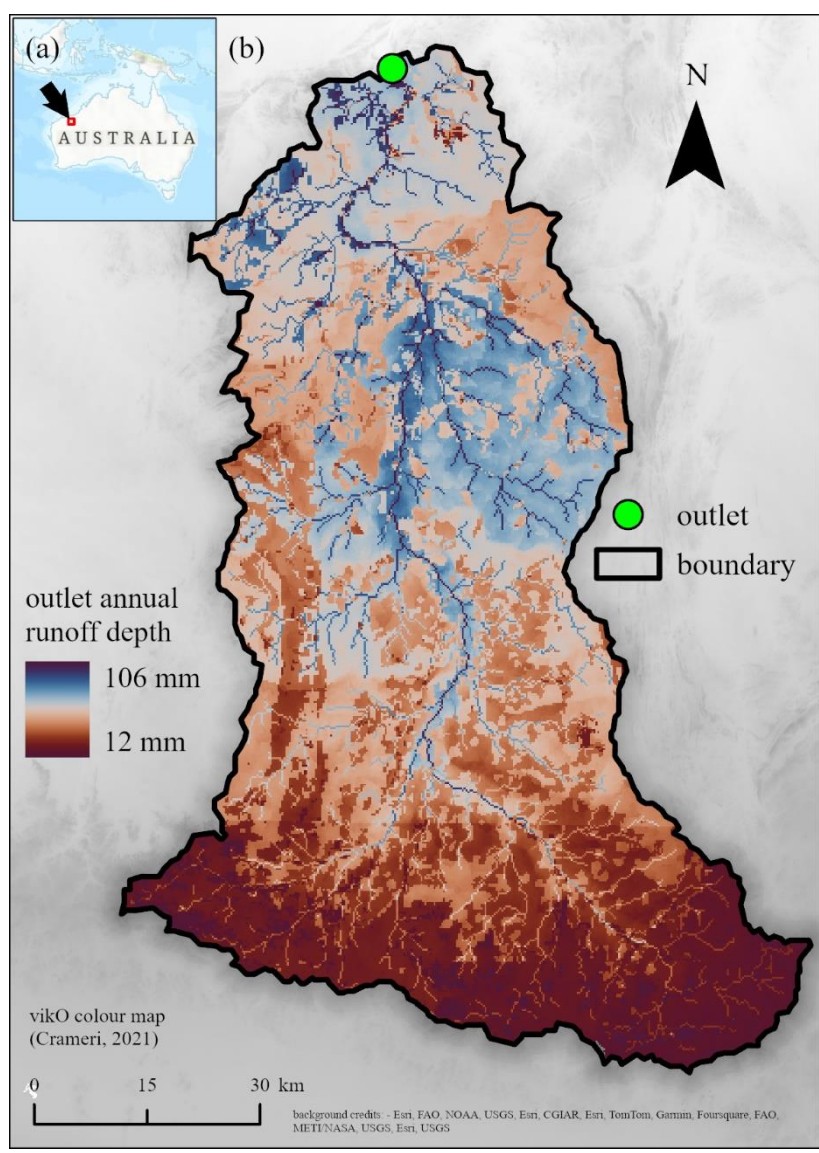

**Figure 3: (a) location of AUNP catchment; (b) runoff connectivity map (RCM) for the catchment of a gauging station located on the Shaw River, showing mean annual runoff reaching catchment outlet taking transmission losses into account, created using validation data.**





## 3.2. Assessment of model performance

**Table 2** summarises the optimum NSE and Pbias values for each catchment in the calibration and validation stages, for both

the runoff-only ('ro') and runoff-and-transferral-ratio ('tr') versions of the model.

**Table 2**. Values of the Nash-Sutcliffe efficiency (NSE) and percentage of average tendency (Pbias) for calibration (cal) and validation (val) runs of the runoff-only (ro) and runoff-and-transferral-ratio (tr) versions of the model, for each of the 28 test catchments. The first two letters of the catchment code indicate country: Australia (AU), Brazil (BR), Israel (IR), USA (US) or South Africa (ZA).


| catchment code | calibration | | | | validation | | | |
|---|---|---|---|---|---|---|---|---|
| | $NSE_{ro,cal}$ (-) | $NSE_{tr,cal}$ (-) | $Pbias_{ro,cal}$ (%) | $Pbias_{tr,cal}$ (%) | $NSE_{ro,val}$ (-) | $NSE_{tr,val}$ (-) | $Pbias_{ro,val}$ (%) | $Pbias_{tr,val}$ (%) |
| AUFR | -305.564 | 0.039 | 1158.3 | -35.0 | -185.942 | -0.209 | 1382.6 | -3.2 |
| AUGD | -1.966 | 0.705 | 130.0 | 20.0 | 0.668 | 0.609 | 42.9 | -26.7 |
| AULT | -92.259 | 0.105 | 1064.0 | 16.5 | -1.310 | 0.058 | 324.9 | -62.9 |
| AUMF | -0.629 | -0.835 | -52.8 | -63.4 | 0.193 | 0.009 | -34.3 | -48.7 |
| AUMS | -8.442 | 0.240 | 199.0 | 7.4 | -0.407 | 0.672 | 136.7 | -13.1 |
| AUNP | -0.903 | 0.353 | 183.8 | -6.5 | -8.376 | 0.410 | 312.6 | 56.0 |
| AUSJ | -0.327 | -0.407 | -51.8 | -54.4 | -0.764 | -1.875 | -35.5 | -38.8 |
| BRMN | 0.281 | 0.265 | 12.7 | -17.9 | 0.283 | 0.199 | -2.7 | -28.9 |
| BRPR | -5.280 | 0.648 | 206.9 | -3.9 | -0.964 | 0.328 | 228.0 | 6.4 |
| ILOB | -3.110 | -3.155 | -96.5 | -97.1 | -5.055 | -5.138 | -96.4 | -97.0 |
| USMH | -4.046 | -6.209 | -67.6 | -82.2 | -1.899 | -2.092 | -64.0 | -66.7 |
| USNP | -8.220 | -8.501 | -57.0 | -62.8 | -12.502 | -10.082 | -10.8 | -22.0 |
| USSC | -4.646 | -4.688 | -97.3 | -97.7 | -6.489 | -6.540 | -97.6 | -98.0 |
| ZAAN | -0.440 | -0.663 | -45.6 | -50.5 | -0.027 | -0.109 | -35.5 | -41.2 |
| ZABT | -8.615 | 0.044 | 314.1 | -11.7 | -30.761 | -0.387 | 388.9 | 26.4 |
| ZADE | -31.513 | 0.186 | 1175.7 | 4.5 | -37.383 | 0.125 | 1006.1 | -22.4 |
| ZADK | -0.082 | 0.129 | 10.6 | -16.8 | -3.608 | -1.415 | 108.5 | 64.0 |
| ZAHE | -1.467 | 0.129 | 121.2 | -27.9 | -14.182 | -1.712 | 328.0 | 101.7 |
| ZAHH | 0.077 | 0.100 | 4.7 | -18.4 | 0.014 | 0.011 | -23.3 | -42.0 |
| ZAKK | -20.469 | -0.404 | 331.3 | -12.6 | -18.204 | -0.117 | 283.7 | -24.3 |
| ZAMB | -1.951 | -1.972 | -88.9 | -90.3 | -0.882 | -0.926 | -91.3 | -92.4 |
| ZAMK | 0.096 | 0.057 | -44.1 | -47.5 | -0.036 | -0.052 | -11.9 | -17.1 |
| ZAO | -0.866 | -0.061 | 76.6 | 24.6 | -6.524 | -1.319 | 165.1 | 64.7 |
| ZAOS | -28.792 | -0.274 | 527.9 | -4.1 | -0.353 | -0.442 | 61.3 | -73.8 |
| ZASD | -1.661 | -1.890 | -61.9 | -66.6 | -1.512 | -1.755 | -56.0 | -64.9 |
| ZAT | -50.187 | 0.186 | 961.6 | -20.5 | -301.272 | 0.148 | 1646.3 | 43.8 |



| | | | | | | | |
|------|--------|-------|------|-----|--------|--------|------|
| ZAUL | -0.546 | 0.012 | 59.9 | 4.0 | -0.870 | 0.441 | 77.1 | 14.4 |
| ZAW | -0.378 | 0.527 | 62.3 | 2.0 | -3.170 | -0.423 | 87.5 | 19.9 |

A first assessment of these results suggests that the use of transferral ratios improves the performance of the model (i.e., moves the NSE closer to 1 and Pbias closer to zero) as demonstrated in **Table 3**. However, it also indicates that there is a significant proportion of the 28 test catchments used where it does not lead to satisfactory performance by any of the

measures.

**Table 3**. Number (out of 28) catchments which show satisfactory values (NSE > 0; -50 % < Pbias < 50 %) for the NSE and Pbias performance metrics at the calibration (cal) and validation (val) stage of model development.

| performance measure | number of satisfactory catchments runoff only | number of satisfactory catchments with transferral ratio | improved (yes/no) |
|---------------------|-----------------------------------------------|----------------------------------------------------------|-------------------|
| $NSE_{cal}$ | 3 | 16 | yes |
| $Pbias_{cal}$ | 5 | 19 | yes |
| $NSE_{val}$ | 4 | 11 | yes |
| $Pbias_{val}$ | 8 | 17 | yes |

The sensitivity of the NSE to the transferral ratio equation parameters ($k_n$, $p_n$, $k_o$, $p_o$ – see equations (4) and (10)) that were adjusted calibrate the model is shown in **Figure 4**, for an example test catchment (code AULT), which shows typical behaviour for this analysis. It indicates that the NSE was most sensitive to the stream network flow travel time constant, and least sensitive to the overland flow travel time constant, and had intermediate sensitivity to the two curve number power law indices.






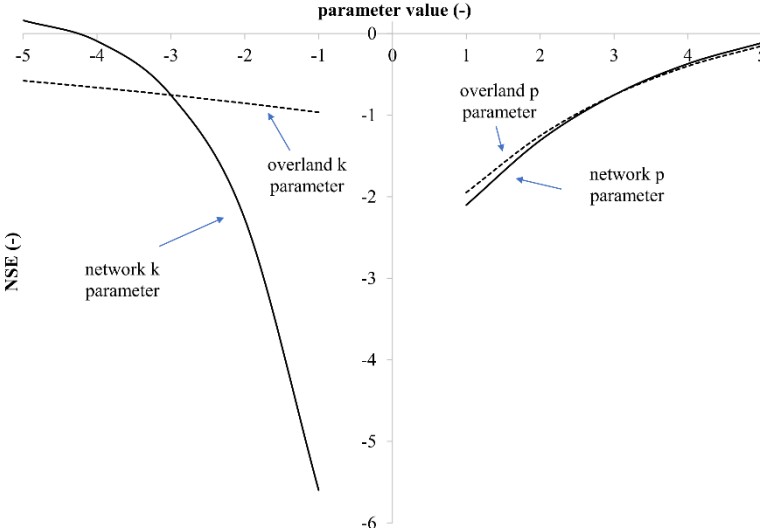

**Figure 4: Sensitivity of Nash-Sutcliffe efficiency (NSE) to variation in transmission loss parameters (catchment code AULT) for calibration data.**

### 3.3. Determination of catchment types for best model performance

To determine which types of catchments the model performs best upon we focus on the results from the validation stage of the modelling (since this is a test of model performance, c.f. the calibration stage, which is an exercise in optimising the model performance) and on the results for model runs where transferral ratios are incorporated (since this is the main novel contribution of this work). On this basis, 11 of the 28 catchments have satisfactory NSE values (>0) and 17 have satisfactory Pbias values (absolute values <50 %), with 9 having both. These nine are listed in **Table 4**, with their predominant land cover characteristics.

**Table 4**. Catchments with NSE > 0 and absolute Pbias values <50 % for validation models with transferral ratio effects incorporated, with the predominant land cover characteristics.

| catchment code | predominant land cover (classification and percentage of coverage) |
|---|---|
| AUGD | grasslands with sparse shrubs 77 % |
| AUMS | closed shrublands 80 % |
| BRMN | mosaic agriculture / degraded vegetation 34 %; open deciduous forest 27 %; Montane forests 500–1000 m - open deciduous 15 % |
| BRPR | open shrublands 32 %; Montane forests 500–1000 m - open deciduous 18 %; Montane forests 500–1000 m - open semi-humid 10 %; mosaic agriculture / degraded vegetation 9 %; agriculture – intensive 8 % |





| | |
|---|---|
| ZADE | open grassland with sparse shrubs 85 % |
| ZAHH | open grassland 47 %; open grassland with sparse shrubs 38 % |
| ZAT | open grassland with sparse shrubs 63 %; closed grassland 34 % |
| ZAUL | deciduous woodland 40 %; croplands (>50 %) 32 % |


**Table 5**. Proportion of catchments of different dominant landcover type for which the model performed satisfactorily in the validation
stage with transferral ratios incorporated.

| dominant landcover type | total number of catchments / 28 | # of catchments with NSE>0; -50 %<Pbias<50 % | % of total catchments with NSE>0; -50 %<Pbias<50 % |
|---|---|---|---|
| grassland | 4 | 4 | 100 |
| agriculture | 1 | 1 | 100 |
| woodland | 2 | 1 | 50 |
| shrubland | 9 | 3 | 33 |
| croplands | 7 | 0 | 0 |
| forest | 5 | 0 | 0 |
| | 28 | 9 | |

Their ranges of values, in the context of the ranges for all 28 catchments, for seven other biophysical catchment parameters
are shown in **Figure 5**. The full set of values of these parameters are provided in tabular format in the **Supplementary
Materials**. The nine catchments with a positive NSE value and absolute Pbias values less than 50 % (**Table 4**) are compared
to all 28 catchments is a summarised format (**Table 5**). **Table 5** shows that while seven of the 28 catchments have a
dominant landcover type of 'croplands' none of these catchments are amongst of the nine catchments that demonstrated
superior model performance (**Table 4**). The same situation is repeated for five catchments with a dominant landcover type of
'forest'. All four 'grassland' catchments (**Table 5**) provided model results that fell into the top nine catchments (**Table 4**) as
did the single catchment with a dominant landcover type of agriculture. Taken as a whole, this suggests that the model works
best in catchments where there is low-growing vegetation (grassland or (pastoral) agriculture), and not well in catchments
where the vegetation is tall and/or dense (croplands and forests).





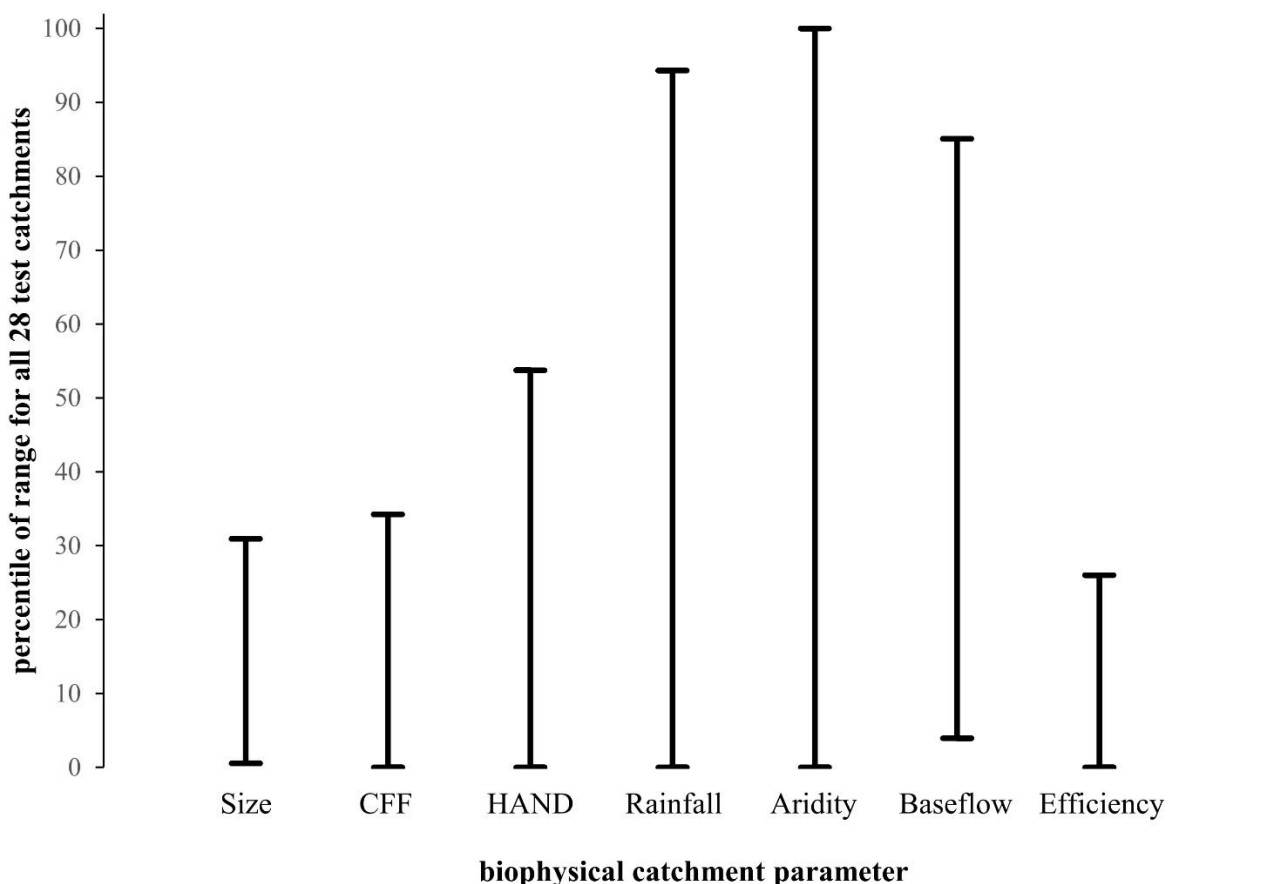

**Figure 5: Range of values of seven biophysical catchment parameters for 9 catchments with NSE > 0 and absolute Pbias < 50 %, in**
**the context of the full range of values for all 28 catchments for each parameter. Size: catchment area; CFF: catchment form**
**factor; HAND: heigh above nearest drainage; Rainfall: mean annual rainfall; Aridity: mean aridity index; Baseflow: % of runoff**
**from baseflow; Efficiency: runoff efficiency.**

Turning next to **Figure 5**, the group of nine catchments for which the model performed satisfactorily are distinguished
amongst the full set of 28 by their (a) relatively small size (<17,027 km$^2$); (b) relatively low (<0.147) catchment form factor
(i.e., lack of elongation in any direction); (c) low (<12.4 %) runoff efficiency (proportion of rainfall that becomes runoff at
the catchment outlet); and to a lesser extent (d) height (<90 m) above nearest drainage (i.e., mean elevation). The appearance
of runoff efficiency in this list suggests the effect of our transferral ratio calculation method being incorporated into the
model. The hydrological parameters considered – mean rainfall, aridity index and percentage of runoff from baseflow, do not
distinguish the nine catchments from the full set of 28 at all. Considered together, therefore, the results shown in **Table 4**,
**Table 5** and **Figure 5** suggest that the model works best in catchments that are relatively small; of approximately equal
length in all directions; of relatively low topography; have high levels of transmission losses of runoff on its journey to the
catchment outlet; and have predominantly low-growing vegetation.



## 4. Discussion

### 4.1 Methodology

HRRTLE employs three global datasets, one being the SRTM void filled elevation product, encompassing 80 % of the Earth's land surface between 60° north and 56° south. This dataset offers a spatial resolution of 3 arc-seconds, approximately 90 m spatial resolution, based on radar data acquired in 2000. The suitability of the elevation dataset for the HRRTLE process relies on the assumption that the elevation data from the year 2000 adequately represents in-stream networks for all modelled years, disregarding changes in morphology. The SRTM void filled elevation data, has previously been applied in hydrologic analyses for water harvesting studies (Sreedevi et al., 2009; Grum et al., 2016; Salih and Hamid, 2017; Ahmed and Diab, 2018; Abdekareem et al., 2022). The HRRTLE process reduces the spatial resolution of the SRTM dataset to 250 m to match the curve number dataset. Consequently, some degree of diminished hydrological model performance might be anticipated compared to using the process with the original 90 m × 90 m data. Yang et al. (2001) investigated the sensitivity of hydrological models to spatial resolution changes, exploring resolutions up to 1,000 m × 1,000 m. They found that the hydrological response is sensitive to changes in the spatial resolution of the DEM, but the significance is greater for hourly response over daily response. Since the HRRTLE process employs a daily temporal resolution for runoff computation then this would suggest that the downgrade in spatial resolution from 90 m to 250 m is not expected to significantly affect runoff computation. Nevertheless, it should be noted that the HRRTLE process goes beyond mere direct runoff computation at individual cells or pixels. Instead, it models the runoff path to the designated catchment outlet and, for the modelled pathways, acquires curve number data to predict transmission losses. Hence, when a DEM is not adequately suited for modelling in-stream networks, there is a greater probability that the curve number data acquired through the HRRTLE process for cells identified as part of the 'network' does not accurately reflect channelised flow.

The HRRTLE modelling process consists essentially of two components. The first component utilises the GPCC precipitation data to predict runoff from each 250 m × 250 m gridded cell using curve number maps (Jaafar et al., 2019) based on the SCS-CN method. The second (novel) component predicts the amount of runoff reaching the catchment outlet from the cell where runoff occurs, accounting for transmission losses. The journey from the runoff cell to the outlet, is modelled as 'overland' flow followed by in-steam 'network' flow. This process makes a universal assumption that, regardless of actual catchment characteristics, overland flow transitions to network flow when the catchment area reaches approximately 4 km$^2$. One of the simplifying assumptions in the HRRTLE process involves maintaining a consistent relationship between flow accumulation (and consequently catchment area) and stream width across the entire catchment. The HRRTLE process assumes that curve number values along the overland and network flowpaths can be used to infer channel properties (i.e., Manning's roughness coefficient). To our knowledge, HRRTLE is the first rainfall-runoff model that utilises such an assumption, although Soni et al. (2022) did estimate equivalent runoff coefficients based on the colour of Google Earth pixels to predict runoff. Associating curve number with hydraulic roughness relies on the presumption that





flowpaths characterised by a higher average curve number are prone to greater wetness or saturation compared to those with lower curve number values. Consequently, these paths pose reduced resistance to open channel flow and, simultaneously, result in fewer transmission losses due to the relatively higher saturation of the ground. Employing curve number values in this manner eliminates the necessity of incorporating extra datasets to define hydraulic roughness. The curve number dataset, utilised for calculating transmission losses along flowpaths, is the same dataset employed in the initial component of the model for runoff computation using the SCN-CN method. As a result, this approach restricts the number of datasets, each carrying its own uncertainties, to three.

**4.2 Performance**

Out of the 28 catchments simulated with HRRTLE, nine demonstrate a positive NSE value and a Pbias within the range of -50 % to 50 % during the validation stage. Additionally, by expanding the Pbias to ±65 %, eleven catchments show a positive NSE value in the validation results. It is thus worthwhile to examine the potential factors contributing to the superior performance of HRRTLE in certain situations and its poorer performance in others. The findings suggest that HRRTLE exhibits improved performance with smaller catchment sizes, although some larger catchments still produced reasonable results. Among the ten catchments categorised as the largest in size, four (AUGD; AUNP; ZADE; ZAT) yielded a positive NSE value during the calibration stage. One plausible explanation could be that the contributing factor to suboptimal results may not be the sheer size of the catchment, but rather the potential for larger catchments to be more diverse and complex. This complexity, which may include engineered structures, could pose challenges for HRRTLE in achieving satisfactory modelling results. None of three large catchments located in North America (USMH; USNP; USSC) gave a positive NSE value for the validation stage.

Certain catchments exhibit relatively high runoff efficiencies, as determined through the calculation using GPCC precipitation and observed discharge data. For instance, one catchment (ILOB) has a runoff efficiency value of 47.7 %. One possible explanation for this is that the observed discharge data incorporates flows beyond those generated solely by precipitation within the catchment boundary — suggesting potential external introduction of water in some manner. Six of the 28 catchments tested in this study had a runoff efficiency greater than 16 %, none of which produced a positive NSE value in the validation stage.

The results suggest a tendency of superior model performance for catchments characterized by low-lying vegetation. One potential rationale is that the radar technology employed to generate the SRTM product encounters difficulties in penetrating vegetation. As a result, digital elevation models (DEMs) for catchments with sparse or low-lying vegetation might be more advantageous for hydrological modelling processes compared to catchments with dense canopy cover. For this study HRRTLE was used to analyse 28 catchments, five of which can be classified by a dominant 'forest' cover yet none of these





five catchments are included in the group of nine catchments that performed better than others in terms of NSE and absolute Pbias (**Table 5**).

Several test catchments (coded as AUFR, ZADE, ZAT) exhibited Pbias values exceeding 1,000 % during the validation stage (**Table 2**) when transmission losses were not taken into account. The calculated baseflows for these catchments are 3.8

%, 7.1 %, and 14.1 %, respectively (see **Supplementary Materials**) which aligns closely with existing literature on this topic. Pilgrim et al. (1988) proposed that the rainfall-runoff modelling approach for arid and semi-arid regions may differ from that for humid zones, as baseflow is essentially absent in arid zone hydrology, while channel transmission losses are crucial. Transmission losses have been documented to surpass 75 % in arid regions (Knighton and Nanson, 1994) and up to ~70 % in semi-arid regions (Abboushi et al., 2015). Studies based in arid and semi-arid regions have generated runoff maps

without incorporating transmission losses (Al-Ghobari et al., 2020; Karimi and Zeinivand 2021; Alataway, 2023; Radwan and Alazba, 2023). Such maps, if used to compute inflows for potential ex-situ water harvesting structures, run the risk of overestimating runoff (or runoff potential) as they do not allow for transmission losses which in dryland regions can be considerable as previously noted. Sayl et al. (2019) created a runoff volume map linking infiltration losses with drainage frequency density. The concept of a connectivity map is not new, as D'Haen et al. (2013) previously developed a

connectivity map of geomorphic coupling for points within a catchment in relation to the catchment outlet. Current rainfall-runoff models exhibit limitations in certain aspects. According to Shanafield et al. (2021, p. 12), rainfall-runoff models "…tell us little of the physical processes and dominant hydrologic flowpaths by which water migrates from its landing place within the catchment to become streamflow." The HRRTLE process addresses this criticism in part, as the connectivity runoff map (e.g., **Figure 3**) provides information on flowpaths and quantifies (in mm) the annual runoff reaching the

catchment outlet (which theoretically could be a potential water harvesting location) for every pixel at a high spatial resolution allowing planners to understand the relative significance of various parts of the catchment with respect to outlet discharge.

**4.3 Application**

The development of the HRRTLE process was geared towards aiding the assessment of potential water harvesting sites within a designated area. This involves a comprehensive examination of numerous locations to pinpoint areas where water harvesting offers the most advantages. The primary phase where HRRTLE is anticipated to deliver valuable applications is during the initial scoping phase of a project cycle. Several characteristics of HRRTLE make it attractive to planners and specialists in the water harvesting sector. Firstly, it relies on freely available global datasets, eliminating the costs and delays

associated with acquiring national data. Secondly, it utilises commonly available software such as MATLAB and ArcGIS Pro, as opposed to specialised hydrologic modelling software. Thirdly, HRRTLE is a relatively straightforward in terms of model construction, requiring only the catchment output and the extent boundaries to be defined by the modeller.





Numerous rainfall-runoff models exist, and some of them have been applied in water harvesting research. As previously
mentioned in this paper, the SCS-CN model is frequently utilised in such studies (Ramakrishnan et al., 2009; Kadam et al.,
2012; Moawad, 2013; Mahmoud, 2014; Pathak et al., 2020; Aghad, 2021; Manaouch et al., 2022). However, none of these
studies address transmission losses. In contrast, the HRRTLE process, while also employing the SCS-CN method, takes
transmission losses into account. The well-known Hydrologic Modelling System software (HEC-HMS) has also been
utilised by researchers in the field of water harvesting (El Osta et al., 2021; Ghanem et al., 2021; Ndeketeya and Dundu,
2021; Ramadan et al., 2022; Soomro et al., 2022) and offers hydrologic modelling features such as runoff hydrographs,
something that the HRRTLE process does not. The Soil and Water Assessment Tool (SWAT) model has also been used in
water harvesting studies, as demonstrated by Ouessar et al. (2009), Al-wadaey et al. (2016), Doulabian et al. (2021) and
Umugwaneza et al. (2022). Ouessar et al. (2009) expressed a preference for a cell-based routing procedure over SWAT's
semi-distributed approach at the subbasin level when modelling flows in arid environments. While the HRRTLE process
does not route hydrographs it does provide a runoff connectivity map at cell-level.

There are various ways in which HRRTLE could be employed in the context of water harvesting site suitability. If a specific
location has been identified for siting a water harvesting structure, HRRTLE could be configured with the catchment outlet
designated as the proposed structure's location. To provide planners with a diverse set of information including a high spatial
resolution runoff connectivity map (e.g., **Figure 3**) which would enable planners to visualise parts of the catchment that
contribute varying amounts of runoff to the outlet. The tool uses almost 40 years of precipitation data allowing the return
period of annual discharge to be determined.

Assuming a consistent relationship between stream width and flow accumulation, the HRRTLE process could be automated
for multiple points, each representing an outlet (i.e., a potential location for a water harvesting structure), by repeating the
transferral ratio elements of HRRTLE (**Figure 1**) without the need to re-process the SCS-CN computations over the 'extent'.
Should the outlets points be sufficiently varied in terms of location and catchment area it would be possible to establish a
regression relationship between catchment area and annual runoff discharge, allowing a prediction of annual runoff for every
pixel within an area of interest. HRRTLE does not route hydrographs, resulting in the absence of peak flow estimations.
Estimating peak inflows in response to a single flood event is essential when designing a water harvesting structure so excess
inflows can bypass safely. Here we suggest that during the initial scoping phase of water harvesting sites, considering the
ratio of annual runoff to the size of the water harvesting reservoir can aid the selection process. This approach would help
streamline site options, allowing modelling efforts for individual events to focus on a reduced number of potential water
harvesting sites. Potentially HRRTLE results could be combined with automated tools that compute water harvesting
reservoir metrics (Petheram et al., 2017; Wimmer et al., 2019; Teschemacher et al., 2020; Delaney et al., 2022) so the site
selection criteria such as reservoir storage volume to annual inflow volume ratio could be computed. The HRRTLE process





does not serve as a substitute for established rainfall-runoff models (such as SWAT and HEC-HMS), but such models could be utilised after HRRTLE has been used to narrow down the number of possible sites or to reinforce the findings from the HRRTLE process.


In cases where observed runoff data is available, HRRTLE can undergo calibration against this data. However, in practical terms, water harvesting scoping studies often occur in regions where obtaining observed runoff data is unavailable (i.e., ungauged catchments). The absence of observed discharge data presents challenges. As Beven (2012, p. 329) notes, "One of the great unsolved challenges in hydrology is the accurate simulation of a catchment without any observational data with

which to calibrate a hydrological model, i.e., an ungauged basin." Despite this obstacle, HRRTLE does possess potential uses in ungauged catchments. Results could be sensitivity-based, with calibration performed for a range of runoff efficiencies. Alternatively, by matching catchment characteristics from gauged catchments where HRRTLE has been successfully calibrated and verified with observed discharge data to another ungauged catchment with the same essential characteristics, HRRTLE parameters could be applied to the ungauged catchment.


## 4.4 Future work

The GPCC precipitation dataset (Ziese et al., 2022) used in the HRRTLE process has a temporal range of 39 years (1982–2020) and a spatial resolution of 1.0 degree × 1.0 degree. The dataset is based on precipitation data provided by national meteorological and hydrological services, regional and global data collections as well as the World Meteorological

Organization Global Telecommunication System data. GPCC offers full global coverage of the Earth's surface. Basheer and Elagib (2019) evaluated this for the Nile Basin, along with another five long-term rainfall products, and ranked the GPCC project the best performing based on monthly, maximum monthly and annual scales. Nevertheless, the spatial resolution of this dataset is rather coarse in comparison to the sizes of most of the 28 catchments examined in this study. To illustrate this point, a precipitation tile with a spatial resolution of 1.0 degree (~110 km on the equator) encompasses an area of about

12,000 km², whereas only 5 of the catchments studied exceed this size. Therefore, future efforts could explore the sensitivity of HRRTLE to precipitation from higher resolution datasets.

In this investigation, the SCS-CN component of the HRRTLE process incorporated an initial abstraction ratio of 0.2, aligning with earlier studies (Sekar and Randhir, 2007; Elewa et al., 2012; Shalamzari et al., 2019). Nevertheless, other

studies have utilised different ratio values (Liu et al., 2021; Weerasinghe et al., 2011). Subsequent research could investigate the consequences of altering the initial abstraction ratio. In this version of HRRTLE the SCS-CN component assumes a permanent dormant season so additional work could investigate how to distinguish between the dormant and growing season especially if this can be achieved without the need to introduce additional datasets. The HRRTLE modelling process computes annual runoff based on the calendar year, presuming that the entirety of annual discharge stems exclusively from





precipitation occurring within the same calendar year. Future work could explore the effects of re-designing the modelling process, so the annual discharge calculations begin at the end of the dry season for example.

To model the 28 catchments using the HRRTLE process and establish the relationship between stream width and flow accumulation, a manual and subjective approach was employed, relying on individual judgment to delineate the riverbank.

Introducing an objective and automated process to determine the relationship between stream width and flow accumulation would likely be advantageous. Flow width rasters can be created using SAGA-GIS (Conrad, 2009) based on the work by Gruber and Peckham (2009) and Quinn et al. (1991) yet flow widths are limited by the spatial resolution of the raster pixels. Automated tools based on remote sensing products have been developed such as RivWidthCloud (Yang et al., 2020) and GrabRiver (Wang et al., 2022), while Mengen et al., 2020 created an automated process using Sentinel-1 products. These

tools and processes could potentially automate the stream width measurement stage in future iterations of HRRTLE. However, the effectiveness of these tools to measure river widths for non-perennial river systems (typically associated with water harvesting) may be diminished compared to perennial rivers.

The HRRTLE process utilises in-stream network cells to extract CN values as a component of its methodology for

calculating the transferral ratio. It is preferable therefore for an accurate alignment between the modelled in-stream cells and the actual stream network. The underlying assumption is that the current version of HRRTLE adequately maps genuine in-stream networks. Nonetheless, future work could assess the sensitivity of network modelling accuracy and explore ways to enhance the modelling process by using other elevation products than the SRTM void filled used in this study. HydroSHEDS is modified SRTM elevation data that integrates hydrographic baseline data (Lehner and Grill, 2013), is freely available, and

could be more appropriate for the HRRTLE process should it offer superior modelling of stream networks so allowing the extraction of more pertinent values from the Curve Number dataset. Alternatively, it is possible to transform a Digital Surface Model (DSM) to a Digital Terrain Model (DTM) by eliminating forest canopy height so making it more preferrable for hydrologic modelling performance. Such a procedure has been carried out on a Copernicus DEM with a land cover dataset (Potapov et al., 2021) creating a DTM dataset (Strahlendorff, 2024).


The HRRTLE process uses gridded maps of curve numbers (Jaafar et al., 2019) the development of which was based on the USDA Soil Conservation Service (SCS) Runoff Curve Number (CN) method. Sujud and Jaafar (2022) conducted runoff computations using this dataset in conjunction with the SCS-CN method, revealing that model performance was influenced by factors such as climate, soil permeability, and bedrock permeability. A potential area for future investigation could

examine of the influence of permeability, particularly bedrock permeability, on the accuracy of HRRTLE results.

In arid and semi-arid regions, the primary source of groundwater recharge is often considered to be transmission losses in ephemeral river systems (Shanafield et al., 2021). Although HRRTLE does not explicitly distinguish between losses

attributed to evapotranspiration and those due to infiltration, in certain catchments, infiltration (representing groundwater
recharge) can account for as much as 95% of all transmission losses (McMahon and Nathan, 2021). Therefore, while
HRRTLE was developed to quantify runoff, there exists the potential for HRRTLE outputs to be utilised to estimate
groundwater recharge.

Model outputs for all catchments underwent verification against observed runoff data, yet acquiring this data from arid and
semi-arid regions, crucial for validating runoff models, poses a considerable challenge. The scarcity of observed data stands
out as a major issue for runoff modelling in arid regions (Pilgrim et al., 1988). While modelling can complement
measurements, it cannot serve as a substitute for them (Silberstein, 2006). The limited availability of observed runoff data in
arid and semi-arid regions, particularly in regions where water harvesting is practiced, impedes the development of rainfall-
runoff models, including the advancement of the HRRTLE process. Therefore, future efforts to enhance discharge
measurement techniques, enabling the verification of models against actual flow data, would be valuable.

## 5 Conclusion

Relying on three global datasets in conjunction with satellite imagery, a rainfall-runoff modelling process has been
developed to compute annual catchment runoff and provide a high spatial resolution runoff connectivity map at 250 m × 250
m. The outcomes of this process, referred to as High Resolution Runoff and Transmission Loss Estimator (HRRTLE) tool,
underwent validation against observed runoff data, achieving satisfactory results in some instances but not universally.
HRRTLE integrates and tests the hypothesis that curve number values can be indicative of the hydraulic properties of surface
flow. It is anticipated that HRRTLE could prove valuable for specialists engaged in water harvesting site selection,
particularly those seeking to employ the SCS-CN method for runoff prediction but wish to account for transmission losses, a
capability offered by the HRRTLE process. Further efforts are recommended to refine and enhance the HRRTLE process
and deepen the understanding of which catchment characteristics are more likely to yield acceptable results. The scarcity of
suitable observed discharge data poses a challenge in the development of rainfall-runoff modelling procedures like
HRRTLE.

## Code availability

The Matlab and ArcPy code to produce runoff connectivity maps is available at DOI: 10.17635/lancaster/researchdata/613.





**Supplement link**

Supplementary Materials are provided.

**Author contributions**

RGD devised the modelling concept, wrote the software code, and prepared the original draft manuscript. GAB, AMF and JDW provided supervision. AMF and JDW reviewed/edited the manuscript.

**Competing interests**

The authors declare that they have no conflict of interest.

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





**Appendices**


**Table A1.** Classification of catchments based on catchment size, form factor, and Height Above Nearest Drainage (HAND).

|  | Class 1 | Class 2 | Class 3 |
|---|---|---|---|
| catchment area (km$^2$) | <= 970 | > 970–<= 6000 | > 6000 |
| form factor (-) | <= 0.105 | > 0.105–<= 0.150 | > 0.150 |
| HAND (m) | <= 35 | > 35–<= 80 | > 80 |

**Table A2.** Summary of catchments with classifications.

| catchment code | extent code | country code | GRDC number | record start (y) | record end (y) | monthly missing values (%) | mean Aridity Index | catchment area class | form factor class | HAND class |
|---|---|---|---|---|---|---|---|---|---|---|
| AUMS | AUS-3 | AU | 5109175 | 1968 | 2019 | 0.0 | 0.37 | 1 | 1 | 1 |
| ZAO | SA-3 | ZA | 1160660 | 1972 | 2021 | 0.0 | 0.48 | 1 | 1 | 2 |
| ZAUL | SA-3 | ZA | 1160704 | 1981 | 2022 | 2.7 | 0.58 | 1 | 1 | 3 |
| ZAKK | SA-1 | ZA | 1160120 | 1964 | 2022 | 2.6 | 0.31 | 1 | 2 | 1 |
| ZAMK | SA-2 | ZA | 1196570 | 1955 | 2022 | 2.3 | 0.33 | 1 | 2 | 2 |
| ZABT | SA-3 | ZA | 1160530 | 1980 | 2022 | 2.0 | 0.30 | 1 | 2 | 3 |
| AULT | AUS-1 | AU | 5606097 | 1978 | 2019 | 0.0 | 0.18 | 1 | 3 | 1 |
| ZAMB | SA-1 | ZA | 1160250 | 1965 | 2022 | 0.9 | 0.28 | 1 | 3 | 2 |
| ZAOS | SA-2 | ZA | 1196561 | 1966 | 2021 | 1.7 | 0.33 | 1 | 3 | 3 |
| AUMF | AUS-1 | AU | 5606042 | 1952 | 2019 | 0.0 | 0.34 | 2 | 1 | 1 |
| BRPR | AS-1 | BR | 3650620 | 1973 | 2020 | 2.8 | 0.52 | 2 | 1 | 2 |
| ZADK | SA-3 | ZA | 1160527 | 1980 | 2022 | 0.6 | 0.25 | 2 | 1 | 3 |
| AUSJ | AUS-1 | AU | 5606040 | 1956 | 2019 | 0.0 | 0.47 | 2 | 2 | 1 |
| ZAHH | SA-1 | ZA | 1159110 | 1927 | 2022 | 1.2 | 0.11 | 2 | 2 | 2 |
| ZAW | SA-2 | ZA | 1197505 | 1968 | 2021 | 0.0 | 0.49 | 2 | 2 | 3 |
| AUFR | AUS-2 | AU | 5607080 | 1967 | 2021 | 11.1 | 0.14 | 2 | 3 | 1 |
| ZAHE | SA-2 | ZA | 1196300 | 1962 | 2022 | 2.4 | 0.28 | 2 | 3 | 2 |
| ILOB | ME-1 | IL | 6594050 | 1970 | 2019 | 0.0 | 0.39 | 2 | 3 | 3 |
| ZAT | SA-2 | ZA | 1159400 | 1923 | 2022 | 3.8 | 0.26 | 3 | 1 | 1 |
| USMH | AN-1 | CA | 4213250 | 1911 | 2021 | 2.6 | 0.46 | 3 | 1 | 2 |
| ZAAN | SA-3 | ZA | 1159650 | 1914 | 2022 | 1.4 | 0.46 | 3 | 1 | 3 |
| AUNP | AUS-2 | AU | 5607520 | 1967 | 2019 | 0.0 | 0.11 | 3 | 2 | 1 |
| ZADE | SA-4 | ZA | 1159305 | 1980 | 2022 | 1.6 | 0.18 | 3 | 2 | 1 |
| USNP | AN-2 | US | 4151514 | 1938 | 2022 | 0.0 | 0.22 | 3 | 2 | 2 |
| ZASD | SA-1 | ZA | 1160305 | 1966 | 2022 | 2.5 | 0.28 | 3 | 2 | 3 |



| | | | | | | | | | |
|---|---|---|---|---|---|---|---|---|---|
| AUGD | AUS-3 | AU | 5109110 | 1969 | 2021 | 0.0 | 0.18 | 3 | 3 | 1 |
| BRMN | AS-1 | BR | 3650634 | 1973 | 2020 | 2.9 | 0.42 | 3 | 3 | 2 |
| USSC | AN-1 | US | 4115220 | 1929 | 2021 | 0.0 | 0.35 | 3 | 3 | 3 |


**Table A3.** Extent references, with name of region, and boundary limits (degrees) for each cardinal direction.

| Reference | Region | West (°) | East (°) | South (°) | North (°) |
|---|---|---|---|---|---|
| AN-1 | Americas–North | -122.0 | -109.0 | 43.0 | 53.0 |
| AN-2 | Americas–North | -107.0 | -103.0 | 34.0 | 37.0 |
| AS-1 | Americas–South | -41.0 | -37.0 | -9.0 | -3.0 |
| ME-1 | Middle East | 34.0 | 37.0 | 32.0 | 35.0 |
| SA-1 | Southern Africa | 18.0 | 22.0 | -35.0 | -31.0 |
| SA-2 | Southern Africa | 24.0 | 32.0 | -29.0 | -23.0 |
| SA-3 | Southern Africa | 24.0 | 32.0 | -34.0 | -28.0 |
| SA-4 | Southern Africa | 22.0 | 27.0 | -33.0 | -28.0 |
| AUS-1 | Australasia | 116.0 | 119.0 | -36.0 | -32.0 |
| AUS-2 | Australasia | 117.0 | 121.0 | -24.0 | -20.0 |
| AUS-3 | Australasia | 137.0 | 146.0 | -21.0 | -17.0 |

**Table A4.** Predominate land cover type for catchments to a minimum of 70 % total coverage.

| catchment code | land cover class name with percentage of coverage |
|---|---|
| AUFR | Closed shrublands 92 % |
| AUGD | Grasslands with sparse shrubs 77 % |
| AULT | Croplands 96 % |
| AUMF | Croplands 56 %; Closed shrublands 14% |
| AUMS | Closed shrublands 80 % |
| AUNP | Open Shrublands   50 %; Grasslands with sparse shrubs 45 % |
| AUSJ | Open forest (Eucalyptus) 42 %; Closed shrublands 24 %; Croplands 14 % |
| BRMN | Mosaic agriculture / degraded vegetation 34 %; Open deciduous forest 27 %; Montane forests 500-1000m - open deciduous 15 % |
| BRPR | Open shrublands 32 %; Montane forests 500-1000m - open deciduous 18 %; Montane forests 500-1000 - open semi-humid 10 %; Mosaic agriculture / degraded vegetation 9 %; Agriculture – intensive 8 % |
| ILOB | Cropland 53 %; Herbaceous, single layer 26 % |
| USSC | Temperate or subpolar needleleaved evergreen forest—closed canopy 58 %; Temperate or |



| | |
|---|---|
| | subpolar mixed broadleaved and needleleaved dwarf-shrubland— open canopy 40 % |
| USMH | Temperate or subpolar needleleaved evergreen forest—closed canopy 41 %; Temperate or subpolar grassland 40 % |
| USNP | Temperate or subpolar needleleaved evergreen forest—closed canopy 40 %; Temperate or subpolar mixed broadleaved and needleleaved dwarf-shrubland— open canopy 39 % |
| ZAAN | Open deciduous shrubland 45 %; Open grassland with sparse shrubs 29 % |
| ZABT | Open deciduous shrubland 49 %; Deciduous woodland 44 % |
| ZADE | Open grassland with sparse shrubs 85 % |
| ZADK | Open deciduous shrubland 67 %; Deciduous woodland 19 % |
| ZAHE | Deciduous woodland 69 %; Croplands (>50%) 26 % |
| ZAHH | Open grassland 47 %; Open grassland with sparse shrubs 38 % |
| ZAKK | Croplands (>50 %) 58 %; Deciduous shrubland with sparse trees 42 % |
| ZAMB | Croplands (>50 %) 63 %; Deciduous shrubland with sparse trees 28 % |
| ZAMK | Croplands (>50 %) 74 % |
| ZAO | Croplands (>50 %) 40 %; Deciduous woodland 26 %; Closed deciduous forest 25 % |
| ZAOS | Croplands (>50 %) 45 %; Deciduous woodland 25 % |
| ZASD | Open deciduous shrubland 30 %; Open grassland with sparse shrubs 30 %; Deciduous woodland 16 % |
| ZAT | Open grassland with sparse shrubs 63 %; Closed grassland 34 % |
| ZAUL | Deciduous woodland 40 %; Croplands (>50 %) 32 % |
| ZAW | Closed deciduous forest 54 %; Deciduous woodland 24 % |