# Peer review of "HRRTLE (High Resolution Runoff and Transmission Loss Estimator): a novel tool for mapping connectivity of runoff in ephemeral stream networks to aid the siting of water harvesting structures"

_Hydrology and Earth System Sciences, 2024_

## Referee Comment (RC1)

Review

The authors propose an approach to account for transmission losses when estimating runoff in arid areas with ephemeral stream networks. They utilize high-resolution datasets and streamflow observations for model calibration and validation. While the methodology appears to improve performance in fewer than half of the analyzed basins, it does not significantly advance our understanding of the underlying processes or contribute to their accurate quantification. Furthermore, the assumptions and limitations of the approach are not sufficiently discussed, which raises concerns about its broader applicability. These critical gaps weaken the robustness of the approach, and I strongly recommend that the authors address these issues to improve the manuscript's quality. The manuscript would also benefit from a clearer structure to better articulate the methodology. Therefore, I recommend that the manuscript, in its current form, should not be considered for publication in the HESS journal.

Major comments

The authors need to clearly define transmission losses, including the specific processes encompassed by this term, and maintain consistency in its usage throughout the document. At present, the definition appears to shift depending on the context. To improve clarity, the methods section should be organized into distinct subsections addressing individual components, such as runoff estimation, datasets used, and evaluation. Each section should explicitly justify the choice of methods or datasets, which would also help eliminate the repetitive content scattered throughout the manuscript.

Moreover, the assumption that all basins contain ephemeral stream networks is critical to the application of the proposed methodology. However, no effort is made to evaluate this assumption. Factors such as water table depth, which could influence transmission losses through ephemeral streambeds, are not considered, and the limitations of this approach are left unaddressed.

The methodology also relies on the use of reduction factors (referred to as 'Transferal ratios'), yet no justification is provided for their application. Additionally, this approach depends on the assumed low uncertainty of Curve Number (CN) values, which are known to be highly uncertain. No attempt is made to assess the potential impact of this uncertainty on the model's performance.

It is unclear how the authors can assume that grid size does not significantly influence model performance. The spatial aggregation of topography alters flow paths and catchment areas, which is highly likely to affect the model's accuracy. A multi-scale analysis is essential to properly address this issue.

Furthermore, MATLAB and ArcGIS are commercial applications with may limit its accessibility. The broader user base would greatly benefit if these tools were developed using free and open-source platforms like Python. Is there a specific reason for not using free software in this case?

Regarding Equation 4, there is no explanation provided for how the authors derived this equation. A reference or a detailed description of the underlying assumptions is necessary for clarity and validation.

Similarly, for Equation 7, why is a linear relationship assumed? What is the accuracy or error associated with this equation? These critical aspects need to be justified for the methodology to be considered robust.

Line 15: Specify which dataset is being used.

Line 17: Clarify what the term 'forecasting' refers to in this context.

Line 97: The statement, "Typically, modelling a catchment to incorporate such transmission losses involves aggregating land into sub-catchments with uniform runoff-loss characteristics." is misleading. This is not typically how transmission losses are estimated.

Line 109: "Create a model to compute generated runoff using global precipitation and curve number datasets" – there are many existing models that do this.

Line 110: "Model flowpaths from points where runoff is generated to the catchment outlet" – all spatially distributed models already perform this function, so why is this presented as a research goal?

Line 112: This is not an appropriate way to evaluate transmission losses.

Line 114: The statement is too vague. Specify which characteristics you are referring to.

Lines 116–117: "The novel contributions of this work lie in the use of fully distributed data sets" – using high-resolution datasets is not novel, as many models are already capable of using them.

Line 118: "In arid and semi-arid regions, there are far fewer rainy days than in humid regions. Only some rainy days create direct runoff. Even fewer rainy days are responsible for runoff reaching a collection point". How are these be assumptions? The authors should clearly state which processes they are attempting to model with this methodology (e.g., infiltration excess, overland flow).

Line 121: "Within such ephemeral systems, baseflow is less significant, or largely absent, compared to more humid regions". How are ephemeral streams defined here?

Line 122: "The method described here exploits these characteristics of arid zone hydrology" – in what way?

Line 123: "generating runoff using daily precipitation data, while surface flow (and hence transmission loss) is modelled as a singular annual event". Why is this approach taken?

Line 124: "Such an approach negates the need to route hydrographs hence sub-basins do not have to be created and catchments can be modelled at relatively high spatial resolution". The meaning of this statement is unclear.

Line 146: What is the difference between an in-stream cell and an overland flow cell? These terms are critical to the methodology and should be clearly defined upfront.

Line 249: "The calculation of the stream network travel time is more complex and is described in the following section". Why is this considered complex?

Line 259: "- more complex –" Is it necessary to highlight this? Is the method really complex, or are the authors referring to computational demands?

Line 517: The term 'flow paths' refers to all routes water may take to reach the basin outlet. This could mislead readers, and the authors do not provide evidence of addressing this issue in the manuscript.

Line 468: "these paths pose reduced resistance to open channel flow and simultaneously, result in fewer transmission losses due to the relatively higher saturation of the ground". This statement is unclear. The streambed of ephemeral streams can become saturated and still lose water, and transmission losses can be influenced by stream stage, due to the hydraulic gradient across the streambed.

Line 471: "this approach restricts the number of datasets, each carrying its own uncertainties, to three". There is no analysis to support this conclusion, and it could be one reason for the model's low performance.

Lines 481–482: "the contributing factor to suboptimal results may not be the sheer size of the catchment, but rather the potential for larger catchments to be more diverse and complex.", Do you mean heterogeneity?

Line 490: "One possible explanation for this is that the observed discharge data incorporates flows beyond those generated solely by precipitation within the catchment boundary", This statement needs more specificity. Are you referring to human interaction?

Line 495: "One potential rationale is that the radar technology employed to generate the SRTM product encounters difficulties in penetrating vegetation". As stated, this does not seem sufficient. How does this impact the calculation, and what about the influence of the CN values?"

---

## Referee Comment (RC2)

**Summary:**

This manuscript presents a novel framework to incorporate transmission losses into existing rainfall runoff models. The authors present a workflow that uses publicly available datasets to calculate hydrometrologic fluxes and watershed structure to quantify the amount of transmission losses. The results presented found that incorporating transmission losses into models have a mixed rate of success (9 out of 28 catchments saw improvement) but showed marked success in those catchments.

**Overarching thoughts:**

First, I want to thank the authors for presenting a framework that highlights the recent advancements and interest in non-perennial systems. Work such as this is important to our fundamental understanding of these systems. Below I summarize some suggestions that could help strengthen the manuscript.

1. Comment: This manuscript is unique in that it uses publicly available and accessible data as inputs into the workflow as well as providing processing code (the DOI provided did not work unfortunately). However, this seems contrary to the processing tools used of ArcGIS and Matlab both of which require expensive licenses to run analysis, and the workflow presented here. While I don't want to disparage the authors on this choice, highlighting freely available datasets in line 529 with the paid nature of the software seems counterintuitive.
Suggested action: I would consider not highlighting the point that the data is freely available.

2. Comment: On the topic of data used, I am curious to why the authors did not use CAMELS/CARVAN datasets that leverage all the needed precipitation, watershed attributes, and land use data needed for the analysis in one common location? I worry that presenting a workflow that leverages many datasets that a user must collect and provide rationale for using, outside the standard for the hydrologic modeling community, might present problems for users as well as produce duplicate tools.
Suggested action: Either a comparative analysis of how the products used here compare to other data sources (i.e. CAMELS) or a rationale why these products were used over other more accessible products.

3. Comment: A theme that perplexes me throughout the manuscript is what is it within the catchments that make the model perform "better" or "worse". Are there spatial patterns? Is it related to a baseflow, groundwater influence, etc.? The relationships of "why" this model performs better don't seem to be well established instead this model behavior as presented now seems to be an emergent behavior. For example,

the authors state on lines 479-480 "…HRRTLE exhibits improved performance with smaller catchment sizes.." and in subsequent paragraphs highlight runoff ratios as potentially important. However, in simply plotting NSE vs these characteristics there seems to be little correlation between goodness-of-fit and these watershed characteristics (see below).

[Figure]

Suggested action: A more rigorous exploratory analysis of model results that include statistical tests (t-test, correlation plots, PCA, etc.) or any additional quantitative analysis that relates model performance to hydrologic and watershed function.

4. Comment: The title of the manuscript uses the word "ephemeral" but the basis of the manuscript is largely focused on arid regions which is not exclusively 1:1 with

ephemeral networks. For example, Brinkerhoff et al. (2024) showed that between 40% and 60% of the river and stream network in the contiguous US is ephemeral with significant portion of ephemeral networks located in humid regions. Additionally, the large-size of some of the watersheds in this study may incorporate majority ephemeral systems, but higher-order streams are analyzed for losses.
Suggested action: I would drop ephemeral from the manuscript where appropriate and replace with arid/semi-arid.

5. Comment: It would be great to know the magnitude of transmission losses predicted in HRRTLE to understand how much streamflow is being lost in these systems, and therefore cannot be captured with water harvesting practices. This could add significant impact to the manuscript.
Suggested action: Calculate transmission loss to streamflow ratio or volume of streamflow lost for catchments.

6. There have been other studies that have looked at spatial/watershed connectivity on a higher resolution or related to climate, physiography, etc.  It would be good to highlight them or at least cite them as they would help bolster the introduction and discussion.
   - Husic et al., 2022: https://doi.org/10.1029/2022GL099898
   - Chen et al., 2019: https://www.nature.com/articles/s41586-019-1558-8

Suggested action: Authors choice.

**Specific feedback:**

Line 334: This is confusing to me. The catchments have streamgages that are used to calculate the runoff ratio? Please clarify.

After figure 3: Larger map (like figure 2) where watershed points are colored by goodness-of-fit metric of choice. This would help a reader discern spatial patterns (if any).

Lines 478-479: Superior compared to what? There were no other instances of models compared, correct? Just incorporation of TL and non-TL simulations?

Line 491: What was the degree of development in the catchments? Comparing how much "disturbance" is in a catchment could lend insight into the varying degrees of runoff efficiency and therefore how important transmission losses may be in a catchment.

Lines 535-545: This paragraph seems disorganized and a bit tough to read. This seems like it would be better as a table or reduced to a single line that states "Studies that utilize varying types of hydrologic models (rainfall-runoff, hydrodynamic, process-based, etc) do not explicitly represent transmission losses (citations)." Then transition to why this is

important tied to the results of this study. Right now, this reads as a "bashing" of other studies.

---

## Author Comment (AC3)

**Author Comments**

AC: We thank the referee for their comments on our manuscript and have proposed ways in which these may be addressed in a resubmission. Our revisions place a stronger emphasis on the context of the study (water harvesting, data-poor locations), hydrological processes (supported by the inclusion of a new figure), and a more rigorous analysis of performance (PCA and subsequent appraisal of catchment characteristics).

**Referee #1**

RC: Review

The authors propose an approach to account for transmission losses when estimating runoff in arid areas with ephemeral stream networks. They utilize high-resolution datasets and streamflow observations for model calibration and validation. While the methodology appears to improve performance in fewer than half of the analyzed basins, it does not significantly advance our understanding of the underlying processes or contribute to their accurate quantification. Furthermore, the assumptions and limitations of the approach are not sufficiently discussed, which raises concerns about its broader applicability. These critical gaps weaken the robustness of the approach, and I strongly recommend that the authors address these issues to improve the manuscript's quality. The manuscript would also benefit from a clearer structure to better articulate the methodology. Therefore, I recommend that the manuscript, in its current form, should not be considered for publication in the HESS journal.

AC: Central to the HRRTLE tool is the aim of developing a relatively simple scoping tool that can support water harvesting studies in data-poor environments. It is not an attempt to advance the understanding of underlying processes, although it does rely on assumptions typical of arid zone hydrology. Elements of the HRRTLE tool's modelling may be described as 'empirical', and such modelling approaches are regarded as the most suitable when attempting to predict hydrological behaviour in ungauged catchments. Globally, large areas can be described as data-poor (e.g., ungauged) and located within arid and semi-arid regions, hence there is significant potential applicability for a tool such as HRRTLE. Model accuracy for data-poor areas will be inferior to models produced in data-rich environments, so the expectations of what is considered acceptable accuracy need to be taken into context and relative to existing methodologies.

Given the range of catchments used to test model results, the authors are not surprised by the variation in model performance. A key issue is gaining insights (objectively) into why some catchments provide better results than others, and suggestions for ways forward are provided later in this document.

The authors accept that the structure of the manuscript can be re-assessed and improvements made. In addition, more can be done to convey the underlying concept and methodology to readers.

RC: Major comments

The authors need to clearly define transmission losses, including the specific processes encompassed by this term, and maintain consistency in its usage throughout the document. At present, the definition appears to shift depending on the context. To improve clarity, the methods section should be organized into distinct subsections addressing individual components, such as runoff estimation, datasets used, and evaluation. Each section should explicitly justify the choice of methods or datasets, which would also help eliminate the repetitive content scattered throughout the manuscript.

AC: In any revised manuscript, the authors aim to clarify the issue of 'transmission losses' by including definitions from previous literature, explaining how transmission losses are defined within the HRRTLE tool processes, and ensuring consistency. With regard to the other points, the authors believe that the responses provided in this document will help address the concerns raised.

RC: Moreover, the assumption that all basins contain ephemeral stream networks is critical to the application of the proposed methodology. However, no effort is made to evaluate this assumption. Factors such as water table depth, which could influence transmission losses through ephemeral streambeds, are not considered, and the limitations of this approach are left unaddressed.

AC: Thank you for this comment. Baseflow has been calculated for all modelled catchments (e.g., Figure 5 on page 19) as an indicator of the degree of ephemerality of streamflow. Therefore, the authors refute the claim that no effort has been made to evaluate this assumption. The authors argue that catchments with minimal baseflow are indicative of high ephemerality. Essentially, HRRTLE is intended as a scoping tool to provide useful information that may inform the selection of suitable sites for water harvesting. We are developing this tool for use in data-poor environments, and as such, many assumptions will need to be made.

The HRRTLE modelling process relies on certain assumptions typical of arid and semi-arid catchment hydrology. For this reason, the catchment selected for modelling can, for the most part, be characterised as arid or semi-arid. The authors acknowledge that the study could be improved in terms of how the conceptualisation of the model is presented to the reader. A conceptual figure, highlighting overland and in-stream flow, similar to the following will be included in any revised manuscript and the text re-worded as appropriate:

[Figure]

***Figure R****Error! No text of specified style in document.****1.*** Conceptualisation of the HRRTLE model: Precipitation data (P) is combined with a pixel-level curve number dataset to compute runoff (q). Runoff is routed to the catchment exit (green dot) as overland flow (o) to the nearest river section (red dot), followed by channelised in-stream flow (i). The model assumes, consistent with arid zone hydrology, that catchment exit flows are predominantly ephemeral, river channels remain above groundwater levels, and transmission losses occur primarily due to dry, porous riverbeds.

To support the HRRTLE modelling concept as being appropriate for arid and semi-arid regions, particularly with respect to groundwater levels and transmission losses, we present the following quotations:

"In general terms it can be expected that perennial streams (often found in cold and temperate climates) are primarily gaining, that is, they receive flows from adjacent groundwater systems; conversely, ephemeral streams (typically found in semi-arid and arid regions) are generally disconnected from groundwater systems and are losing." (McMahon and Nathan, 2021, p. 4).

Reference: McMahon, T.A. and Nathan, R.J. (2021) 'Baseflow and transmission loss: A review', WIREs Water. Hoboken, NJ. Available at: https://doi.org/10.1002/wat2.1527.

"Second, rivers are typically ephemeral (having no permanent flow), so channels lose water through dry, porous beds (transmission losses) because water tables lie well below the channel." (Chen, S.-A. et al., 2019, p. 573).

Reference: Chen, S.-A. et al. (2019) 'Aridity is expressed in river topography globally', Nature, 573(7775), pp. 573–577. Available at: https://doi.org/10.1038/s41586-019-1558-8.

RC: The methodology also relies on the use of reduction factors (referred to as 'Transferal ratios'), yet no justification is provided for their application. Additionally, this approach depends on the assumed low uncertainty of Curve Number (CN) values, which are known to be highly uncertain. No attempt is made to assess the potential impact of this uncertainty on the model's performance.

AC: The use of curve number (CN) values with runoff procedures, such as the SCS-CN calculations, is well-established and particularly well-suited for ungauged catchments. The HRRTLE model calculates runoff using the SCS-CN method and then determines losses, specifically the proportion of runoff that reaches the catchment outlet. The 'transferal ratios' essentially divide the runoff into two components: one representing losses ('transmission losses') and the other representing runoff that reaches the catchment outlet. The CN values used in this study have not been calculated by the authors but instead have been sourced from a widely cited global curve number dataset. Whilst variations in CN datasets have not been explored, model performance has been evaluated using NSE (Nash-Sutcliffe Efficiency) and Pbias.

The HRRTLE tool is intended to support researchers working in drylands, particularly in ungauged catchments, during the scoping stages of a water harvesting (e.g., dam) project cycle. HRRTLE is not designed to compete with more sophisticated runoff models, especially in contexts where observed runoff data is available. The authors acknowledge that in a revised manuscript a detailed explanation of why the model was developed could be provided.

It may be beneficial to study the uncertainties associated with the curve number dataset and other datasets used in the model, such as precipitation and DEM (Digital Elevation Model) data in the future, but that is beyond the scope of the current study.

RC: It is unclear how the authors can assume that grid size does not significantly influence model performance. The spatial aggregation of topography alters flow paths and catchment areas, which is highly likely to affect the model's accuracy. A multi-scale analysis is essential to properly address this issue.

AC: The authors assume that this comment refers to Lines 444–447 in the manuscript:

"Yang et al. (2001) investigated the sensitivity of hydrological models to spatial resolution changes, exploring resolutions up to 1,000 m × 1,000 m. They found that the hydrological response is sensitive to changes in the spatial resolution of the DEM, but the significance is greater for hourly response than for daily response."

Here, the authors are citing a multi-scale analysis conducted by Yang et al. (2001). The HRRTLE process uses daily precipitation data (not hourly), which is why we concluded that the spatial resolution adopted for the HRRTLE modelling process is reasonable and that assessment of sensitivities to the spatial resolution of the DEM is beyond the scope of this study.

However, the authors acknowledge the validity of the referee's comment. HRRTLE relies heavily on accurately delineating streams and rivers, as Manning's n is determined based on curve number values along these flow paths. Therefore, the more accurately the flow paths are defined within the HRRTLE model, the better the model's performance is likely to be.

RC: Furthermore, MATLAB and ArcGIS are commercial applications with may limit its accessibility. The broader user base would greatly benefit if these tools were developed using free and open-source platforms like Python. Is there a specific reason for not using free software in this case?

AC: The authors agree that using free and opensource software could have offered some advantages over MATLAB and ArcGIS. However, MATLAB and ArcGIS were chosen simply because the authors were familiar with both, and they are often regarded as 'industry standard'. Now that the issue with DOI minting has been resolved, the code should be fully accessible to all, allowing researchers to adapt it for use with their preferred software.

RC: Regarding Equation 4, there is no explanation provided for how the authors derived this equation. A reference or a detailed description of the underlying assumptions is necessary for

clarity and validation.

AC: HRRTLE represents a novel modelling process, and thus this equation is not directly derived from any previous work, as the authors are not aware of any similar studies (i.e., using the curve number to estimate losses). Equation 4 provides an explanation of how the transferal ratio ($TR_n$) is calculated. An explanation is included directly below Equation 4 in the manuscript. The authors acknowledge that this explanation could be expanded and improved, particularly in describing how Equation 4 aligns with the arid zone hydrological processes that HRRTLE aims to replicate.

RC: Similarly, for Equation 7, why is a linear relationship assumed? What is the accuracy or error associated with this equation? These critical aspects need to be justified for the methodology to be considered robust.

AC: HRRTLE represents a novel modelling process, and a key aspect of this original work involves relating Manning's roughness to the curve number. The authors found some previous literature suggesting that there may be merit in such an approach, but no formula directly linking Manning's n to the curve number. Therefore, the authors consider a linear relationship to be an appropriate starting point. While acknowledging that there may be accuracy issues or errors associated with this equation, or any other equations or datasets, the model's performance is evaluated using NSE and Pbias.

RC: Line 15: Specify which dataset is being used.

AC: Line 15 is in the Abstract section. Here, we aim to keep the text intelligible to the general reader and in line with the HESS guidance, which can be found under "2. Abstract" at:

https://www.hydrology-and-earth-system-sciences.net/submission.html#manuscriptcomposition

Specifically, we wish to draw attention to the sentence: "Reference citations should not be included in this section, unless urgently required, and abbreviations should not be included without explanations."

Therefore, we do not feel that referencing specific datasets in the Abstract adds value for the general reader, and we do not believe a change to the text is necessary.

RC: Line 17: Clarify what the term 'forecasting' refers to in this context.

AC: Here, the word 'forecasting' could be interchangeable with 'predicting'. The HRRTLE modelling process computes runoff at each pixel (using the Soil Conservation Service Curve Number procedure), see Figure R1 above, but only some of this runoff reaches the catchment outlet. We acknowledge, however, that this sentence could be worded more accurately. The HRRTLE tool calculates the runoff volume reaching the catchment outlet, taking transmission losses into account.

RC: Line 97: The statement, "Typically, modelling a catchment to incorporate such transmission losses involves aggregating land into sub-catchments with uniform runoff- loss characteristics." is misleading. This is not typically how transmission losses are estimated.

AC: The authors feel that the paragraph (Lines 86-101) accurately reflects the points they wish to convey, namely that any modelling process that 'lumps' high-resolution data risks losing variability in catchment characteristics. Perhaps the sentence could be slightly altered to:

"Modelling a catchment to incorporate such transmission losses may involve aggregating land into sub-catchments with uniform runoff-loss characteristics."

RC: Line 109: "Create a model to compute generated runoff using global precipitation and curve number datasets" – there are many existing models that do this.

AC: It would have been helpful if some of these models had been specifically named here. It may be useful to consider that the aim of the study is to achieve all five listed objectives, rather than treating each objective as a novel one in itself. These objectives could perhaps be revised to emphasize that one of them was to create a connectivity map without any reduction in spatial resolution from the global curve number dataset.

RC: Line 110: "Model flowpaths from points where runoff is generated to the catchment outlet" – all spatially distributed models already perform this function, so why is this presented as a research goal?

AC: HRRTLE aims to add value to runoff maps (which do not compute runoff at catchment outlets while accounting for losses), typically used by researchers undertaking water harvesting site selection studies. Perhaps this point needs to be stressed more strongly in the text as it is at present.

RC: Line 112: This is not an appropriate way to evaluate transmission losses.

AC: Runoff maps are frequently used in water harvesting studies. These runoff potential maps are useful but do not consider transmission losses. These runoff maps typically use the SCS-CN procedure for computing runoff. The HRRTLE tool also uses this procedure but includes an additional "bolt-on" extension that accounts for transmission losses. Thus, Line 112 is effectively stating that the study aims to consider both scenarios: SCS-CN runoff and SCS-CN runoff with transmission losses. It is important to emphasize the potential differences when transmission losses are not considered and the impact this has on runoff volumes at the catchment outlet.

Therefore, the authors do not feel this line requires removal, but it could perhaps be reworded to something like: "Evaluate the model results against observed runoff data, including assessing the effect of incorporating transmission losses by comparing the results of model runs with and without them."

RC: Line 114: The statement is too vague. Specify which characteristics you are referring to.

AC: Perhaps the phrasing of this line could be improved by stating 'catchment characteristics' or 'catchment biophysical characteristics'. The authors feel it is important to define this term clearly to restrict the range of characteristics that could ultimately be examined.

RC: Lines 116–117: "The novel contributions of this work lie in the use of fully distributed data sets" – using high-resolution datasets is not novel, as many models are already capable of using them.

AC: The authors are not aware of any catchment models that generate a connectivity map without reducing the spatial resolution of an original dataset with a resolution of 250 m. That said, it may be beneficial to rephrase these lines to more clearly highlight the novel aspects of the HRRTLE tool.

RC: Line 118: "In arid and semi-arid regions, there are far fewer rainy days than in humid regions. Only some rainy days create direct runoff. Even fewer rainy days are responsible for runoff reaching a collection point". How are these be assumptions? The authors should clearly state

which processes they are attempting to model with this methodology (e.g., infiltration excess, overland flow).

AC: The characteristics outlined in these lines are central to the assumptions underpinning the HRRTLE tool, which seeks to leverage features typical of arid-zone hydrology. The key point being conveyed here is that runoff events influencing the catchment outlet occur so infrequently that the HRRTLE tool models transmission losses as a single annual event.

The authors acknowledge that more could be done to better communicate the assumptions of the HRRTLE tool, including the addition of a figure similar to Figure R1 presented above.

RC: Line 121: "Within such ephemeral systems, baseflow is less significant, or largely absent, compared to more humid regions". How are ephemeral streams defined here?

AC: We define ephemeral streams "...*when flows are short and in direct response to precipitation*..." (Shanafield, et al., 2021, p. 2).

Ref: Shanafield, M. et al. (2021) 'An overview of the hydrology of non-perennial rivers and streams', Wiley interdisciplinary reviews. Water, 8(2), pp. e1504-n/a. Available at: https://doi.org/10.1002/wat2.1504.)

RC: Line 122: "The method described here exploits these characteristics of arid zone hydrology" – in what way?

AC: Only a few rainfall events in any given year are responsible for generating runoff, and even fewer events result in runoff that reaches the catchment outlet. The groundwater table remains low throughout the catchment, causing water in streams and rivers to be consistently lost via infiltration. Conversely, rivers are not recharged by high groundwater levels. Most transmission losses occur as a result of infiltration. Due to the short duration of significant runoff events, transmission losses due to evaporation are relatively small, especially when compared to infiltration losses. Infiltration losses depend on the dryness of the riverbed. River flows passing through sections of dry riverbeds (indicated in the modelling process by relatively lower curve number values) will experience higher levels of infiltration losses than those passing through sections that are wet (or less dry). The authors recognise the need to carefully consider how best

to incorporate these points into any revised manuscript.

RC: Line 123: "generating runoff using daily precipitation data, while surface flow (and hence transmission loss) is modelled as a singular annual event". Why is this approach taken?

AC: This approach was adopted for its simplicity and because it leverages a characteristic of arid zone hydrology: the assumption that a small number of rainfall events in any given year account for the majority of the runoff reaching a catchment outlet. The HRRTLE tool is designed to assist specialists working in data-poor environments, where a 'reasonable' estimate of annual runoff volumes would be valuable for assessing the feasibility of potential water harvesting structures.

RC: Line 124: "Such an approach negates the need to route hydrographs hence sub-basins do not have to be created and catchments can be modelled at relatively high spatial resolution". The meaning of this statement is unclear.

AC: Since HRRTLE models surface flow (from the runoff generation pixel to the catchment outlet) in an aggregated manner, detailed time-series analysis of hydrographs (which represent flow over time) is unnecessary. Modelling a hydrograph for each pixel would have been computationally demanding and therefore not a realistic option. By omitting hydrograph routing, the model maintains a spatial resolution equal to that of the global curve number dataset, i.e., 250 m × 250 m. In the event of a revised manuscript the authors would aim to clarify the Line 124 sentence.

RC: Line 146: What is the difference between an in-stream cell and an overland flow cell? These terms are critical to the methodology and should be clearly defined upfront.

AC: The HRRTLE tool models a flow path for every catchment pixel to the catchment outlet. The methodology assumes that runoff (generated at each pixel) exhibits 'overland' flow from the pixel in question to the nearest 'in-stream' pixel. 'Overland' flow follows overland flow characteristics, while 'in-stream' flow exhibits channelized flow; hence, different equations govern the two types of flow behaviour.

The authors acknowledge that the manuscript could be improved with the addition of a new figure, possibly indicating both overland and in-stream flow, along with the catchment outlet and

potentially the groundwater table.

RC: Line 249: "The calculation of the stream network travel time is more complex and is described in the following section". Why is this considered complex?

AC: Since "more complex" is used again just a few lines later (Line 259) then this sentence could be reworded to simply: "The calculation of the stream network travel time is described in the following section."

RC: Line 259: "- more complex –" Is it necessary to highlight this? Is the method really complex, or are the authors referring to computational demands?

AC: Here is using the word 'complex' to describe something "involving a lot of different but related parts". The authors believe that this is an appropriate choice of word given the number of elements used to compute the transferal ratios.

RC: Line 517: The term 'flow paths' refers to all routes water may take to reach the basin outlet. This could mislead readers, and the authors do not provide evidence of addressing this issue in the manuscript.

AC: The HRRTLE tool generates a 'flow path' for each individual pixel within the catchment. It models the flow path starting 'overland', progressing to the nearest 'in-stream' pixel, and ultimately reaching the catchment outlet. Including a schematic-style figure (similar to Figure R1 above) to illustrate the key elements of the concept behind HRRTLE could enhance the reader's understanding of the tool and its functionality, while also addressing concerns that the text may be misleading.

RC: Line 468: "these paths pose reduced resistance to open channel flow and simultaneously, result in fewer transmission losses due to the relatively higher saturation of the ground". This statement is unclear. The streambed of ephemeral streams can become saturated and still lose water, and transmission losses can be influenced by stream stage, due to the hydraulic gradient across the streambed.

AC: HRRTLE is not a detailed rainfall-runoff model compared to many others, as its goal is to

predict annual runoff volumes rather than the runoff response to design events. The question, therefore, is whether the level of modelling detail within the HRRTLE methodology is appropriate to achieve the intended aim.

Yes, the HRRTLE process simplifies many of the actual physical processes that occur – but all models must simplify to some degree. The authors stand by the wording of this sentence, as it supports some of the inherent assumptions about the HRRTLE methodology.

In arid and semi-arid hydrology, riverbeds are often dry for most of the time, so losses into the bed are high, particularly when compared to perennial catchment systems. HRRTLE utilises the curve number value to ascertain the degree of 'wetness' in the river system. In theory, therefore, runoff from one area of the catchment may not reach the catchment outlet, as it passes through parts of the catchment with low curve number values, which indicate dry riverbeds and hence higher levels of transmission losses through infiltration. This mimics the actual hydrological processes of dryland systems. As Chen et al. (2019, p. 573) explain:

"In arid climates, streamflow tends to decrease downstream in all but extreme floods for two main reasons. First, low annual rainfall, limited areal coverage of rainstorms, and short duration of rainfall events generate partial area runoff. This results in a small proportion of basin tributaries contributing streamflow to the mainstem for limited periods of time. Second, rivers are typically ephemeral (having no permanent flow), so channels lose water through dry, porous beds (transmission losses) because water tables lie well below the channel."

Ref: Chen, S.-A. et al. (2019) 'Aridity is expressed in river topography globally', Nature, 573(7775), pp. 573–577. Available at: https://doi.org/10.1038/s41586-019-1558-8.

RC: Line 471: "this approach restricts the number of datasets, each carrying its own uncertainties, to three". There is no analysis to support this conclusion, and it could be one reason for the model's low performance.

AC: Not all of the catchments tested using HRRTLE showed low performance. In the study, 28 catchments were tested, each with a wide range of characteristics. The challenge has been analysing the features that produce better performance.

RC: Lines 481–482: "the contributing factor to suboptimal results may not be the sheer size of the catchment, but rather the potential for larger catchments to be more diverse and complex.", Do you mean heterogeneity?

AC: Yes, the authors consider that this sentence could be reworded to something like:

"One plausible explanation could be that the contributing factor to suboptimal results is not the sheer size of the catchment, but rather the potential for larger catchments to exhibit a greater degree of hydrological heterogeneity."

RC: Line 490: "One possible explanation for this is that the observed discharge data incorporates flows beyond those generated solely by precipitation within the catchment boundary", This statement needs more specificity. Are you referring to human interaction?

AC: Yes, here the authors are inferring that for some catchments there may be some development such as channeling water from a catchment where resource planners consider there is a surplus of water to a catchment where there is a deficit. This sentence could perhaps be improved using terms like 'water transfers' and catchment 'interconnections'. Also, there is a possibility of unsustainable extraction of groundwater.

RC: Line 495: "One potential rationale is that the radar technology employed to generate the SRTM product encounters difficulties in penetrating vegetation". As stated, this does not seem sufficient. How does this impact the calculation, and what about the influence of the CN values?"

AC: HRRTLE models the flow paths from each pixel to the catchment outlet, first by considering 'overland' flow and then by 'in-stream' flow. Utilising a DEM that more accurately models terrain elevation, as opposed to surface elevation (which may reflect the elevation of vegetation), will lead to a more accurate representation of the modelled flow paths.

The influence of CN values has not been investigated. The same can be said for the precipitation dataset. Should future versions of HRRTLE be produced, new datasets could be investigated, especially if 'improved' datasets become available. For example, a DEM with a refined river network (e.g., Hydrosheds version 2, when released), an updated Global Curve Number dataset, and a long-term, daily precipitation dataset with a higher spatial resolution than the GPCC dataset used in the study would all be worth examining. At the time of this research, these datasets were

the best available to the authors and were deemed suitable for the intended purpose of developing a tool to support scoping studies on water harvesting in data-scarce dryland environments.

---

## Author Comment (AC4)

AC: We thank the referee for their comments on our manuscript and have proposed ways in which these may be addressed in a resubmission. Our revisions place a stronger emphasis on the context of the study (water harvesting, data-poor locations), hydrological processes (supported by the inclusion of a new figure), and a more rigorous analysis of performance (PCA and subsequent appraisal of catchment characteristics).

Referee #2

**RC: Summary:**

This manuscript presents a novel framework to incorporate transmission losses into existing rainfall runoff models. The authors present a workflow that uses publicly available datasets to calculate hydrometrologic fluxes and watershed structure to quantify the amount of transmission losses. The results presented found that incorporating transmission losses into models have a mixed rate of success (9 out of 28 catchments saw improvement) but showed marked success in those catchments.

**Overarching thoughts:**

First, I want to thank the authors for presenting a framework that highlights the recent advancements and interest in non-perennial systems. Work such as this is important to our fundamental understanding of these systems. Below I summarize some suggestions that could help strengthen the manuscript.

AC: The authors are pleased that work in this subject area is regarded as important. They believe that research into non-perennial runoff, particularly where observed data is unavailable or inadequate, is crucial, especially given the vast proportion of the Earth's land surface classified as drylands.

RC:

1. Comment: This manuscript is unique in that it uses publicly available and accessible data as inputs into the workflow as well as providing processing code (the DOI provided did not work unfortunately). However, this seems contrary to the processing tools used of ArcGIS and Matlab both of which require expensive licenses to run analysis, and the workflow presented here. While I don't want to disparage the authors on this choice, highlighting freely available datasets in line 529 with the paid nature of the software seems counterintuitive.

Suggested action: I would consider not highlighting the point that the data is freely available.

AC: Apologies for the DOI not working. This was an oversight on the part of the authors, as the DOI had not yet been "minted." This issue has now been resolved, and the DOI should hopefully be functioning correctly.

The authors accept this point regarding the proprietary nature of both the ArcGIS and MATLAB licences, as well as the associated costs. We will certainly consider removing the text about freely available data. Should HRRTLE be developed into newer versions, the authors believe that the use of proprietary software should be replaced with open-source alternatives.

2. Comment: On the topic of data used, I am curious to why the authors did not use CAMELS/CARVAN datasets that leverage all the needed precipitation, watershed attributes, and land use data needed for the analysis in one common location? I worry that presenting a workflow that leverages many datasets that a user must collect and provide rationale for using, outside the standard for the hydrologic modeling community, might present problems for users as well as produce duplicate tools.

Suggested action: Either a comparative analysis of how the products used here compare to other data sources (i.e. CAMELS) or a rationale why these products were used over other more accessible products.

AC: Thank you for this suggestion. The CAMELS/CARVAN datasets were not something the authors were previously aware of, so this is valuable information. Should HRRTLE be developed into newer versions, the CAMELS/CARVAN datasets would certainly be worth considering.

To provide some background on the choice of datasets and the rationale behind the development of HRRTLE as presented: the authors observed that researchers working on water harvesting site selection studies often rely on "runoff maps," which they generate by creating their own land-use layers to produce curve number rasters for runoff computation. This laborious procedure can be streamlined by using an existing global curve number dataset (e.g., Jaafar et al., 2019), which is precisely what the HRRTLE tool does. Since the authors are unfamiliar with the CAMELS/CARVAN datasets, we are uncertain if they include a global curve number dataset or if one could be integrated. If they do, that would be ideal, but it is essential for the HRRTLE tool that a global curve number dataset is available for its operation.

In defense of HRRTLE regarding the use of "many datasets," we do not necessarily accept this characterization. Only three datasets (plus satellite imagery) are required to compute runoff

volume: precipitation, a digital elevation model (DEM), and a global curve number (GCN) dataset. The same GCN dataset is used twice, in both the runoff computation and transmission loss processes, making HRRTLE relatively parsimonious in terms of the number of required datasets.

However, we take your point regarding the need for additional datasets in the context of HRRTLE tool research aimed at evaluating catchments and associating specific catchment types with degrees of model performance. This aspect of the analysis does require numerous datasets, and the CAMELS/CARVAN datasets would indeed be helpful in this regard, as suggested.

RC:

3. Comment: A theme that perplexes me throughout the manuscript is what is it within the catchments that make the model perform "better" or "worse". Are there spatial patterns? Is it related to a baseflow, groundwater influence, etc.? The relationships of "why" this model performs better don't seem to be well established instead this model behavior as presented now seems to be an emergent behavior. For example,

the authors state on lines 479-480 "...HRRTLE exhibits improved performance with smaller catchment sizes.." and in subsequent paragraphs highlight runoff ratios as potentially important. However, in simply plotting NSE vs these characteristics there seems to be little correlation between goodness-of-fit and these watershed characteristics (see below).

[Figure]

Suggested action: A more rigorous exploratory analysis of model results that include statistical tests (t-test, correlation plots, PCA, etc.) or any additional quantitative analysis that relates model performance to hydrologic and watershed function.

AC: The authors appreciate this suggestion and accept that a more rigorous analysis of the model

results is warranted. We propose implementing your suggestion of utilising PCA and applying it to the seven catchment characteristics (see Figure 5 of the manuscript). This analysis might reveal whether there is a cluster of 'good' model results based on the principal components of the seven characteristics.

If no such cluster emerges, the analysis could help identify paired catchments—those that are similar to each other based on the principal components of the seven catchment characteristics. By identifying these 'twinned' catchments, where one demonstrates 'good' model performance and the other 'bad,' differences between the two might become apparent. These differences could potentially be observed by comparing imagery of the two catchments, examining disturbances or obvious differences in land use/land cover. While this approach may be somewhat subjective, it represents an improvement on the existing analysis.

RC:

4. Comment: The title of the manuscript uses the word "ephemeral" but the basis of the manuscript is largely focused on arid regions which is not exclusively 1:1 with

ephemeral networks. For example, Brinkerhoff et al. (2024) showed that between 40% and 60% of the river and stream network in the contiguous US is ephemeral with significant portion of ephemeral networks located in humid regions.

Additionally, the large-size of some of the watersheds in this study may incorporate majority ephemeral systems, but higher-order streams are analyzed for losses.

Suggested action: I would drop ephemeral from the manuscript where appropriate and replace with arid/semi-arid.

AC: Thank you for this observation and suggestion. HRRTLE was developed with the assumption that it would be most suitable for catchments containing ephemeral systems. The authors believe that such catchments are often encountered by planners and researchers seeking suitable sites for water harvesting structures.

The challenge, however, lies in obtaining observed runoff data specific to the type of catchments described. Ideally, we would have had access to runoff data exclusively from ephemeral systems. The absence of this data, coupled with the subsequent testing of HRRTLE on runoff observed in higher-order streams, strengthens the argument for omitting the term 'ephemeral'.

The authors are open to removing the term 'ephemeral' and suggest that 'drylands' may be a more suitable alternative to reflect the range of catchments tested.

RC:

5. Comment: It would be great to know the magnitude of transmission losses predicted in HRRTLE to understand how much streamflow is being lost in these systems, and therefore cannot be captured with water harvesting practices. This could add significant impact to the manuscript.

Suggested action: Calculate transmission loss to streamflow ratio or volume of streamflow lost for catchments.

AC: Thank you for this suggestion. This should be entirely feasible. We know the total runoff generated at each pixel and the runoff volume reaching the catchment outlet, so we can calculate the volume lost due to transmission losses and, as suggested, provide a suitable ratio.

6. There have been other studies that have looked at spatial/watershed connectivity on a higher resolution or related to climate, physiography, etc. It would be good to highlight them or at least cite them as they would help bolster the introduction and discussion.
-                Husic et al., 2022: https://doi.org/10.1029/2022GL099898
-                Chen et al., 2019: https://www.nature.com/articles/s41586-019-1558-8

Suggested action: Authors choice.

AC: Thank you for highlighting these two studies.

Husic et al. (2022) is particularly interesting for several reasons. It addresses how reservoirs significantly reduce longitudinal downstream connectivity. A potential avenue for further exploration, should the results warrant additional analysis, is quantifying the degree of connectivity for each catchment modelled using HRRTLE. The authors of HRRTLE hypothesise that the tool yields better model performance when catchment connectivity is high. An option worth considering is employing the "SedInConnect" software utilised by Husic et al. (2022) and applying it to the HRRTLE catchments.

Chen et al. (2019) is also noteworthy. This study offers further evidence regarding the characteristics of catchments that the authors of HRRTLE believe the tool is particularly well-suited for—namely, those with high losses through dry porous beds (transmission losses) and a water table situated well below the riverbed.

**RC: Specific feedback:**

RC: Line 334: This is confusing to me. The catchments have streamgages that are used to calculate the runoff ratio? Please clarify.

AC: The authors agree that this is confusing and could have been better worded. The development of HRRTLE stems from the authors' ultimate goal of seeing its application, most likely in dryland regions where no observed runoff data are available. For the purposes of developing HRRTLE and verifying its results, we opted to use runoff data

from catchments that are (mostly) located in arid and semi-arid regions.

To clarify, the sentence in question could be improved by stating that the intended purpose of HRRTLE is to serve as a tool for researchers and practitioners working with ungauged catchments. A possible replacement for the sentence could be:

"Here, as we envisage HRRTLE being typically used in conjunction with ungauged catchments, we argue that the criterion for good performance should be somewhat relaxed. Therefore, we consider an absolute value of Pbias less than 50% as indicative of the threshold between adequate and inadequate performance."

RC: After figure 3: Larger map (like figure 2) where watershed points are colored by goodness- of-fit metric of choice. This would help a reader discern spatial patterns (if any).

AC: The authors consider this to be a helpful suggestion. Additionally, it may be valuable to include, in the supplementary materials, a figure showing the best-performing and worst-performing catchments, illustrating the catchment boundaries and associated imagery.

RC: Lines 478-479: Superior compared to what? There were no other instances of models compared, correct? Just incorporation of TL and non-TL simulations?

AC: Here, we are discussing the various HRRTLE model performances across the 28 catchments tested. The "superior" performances, as determined by NSE and Pbias, are considered to be better than others.

Possibly, the sentence in question could be reworded as follows:

"It is therefore valuable to explore the factors that contribute to HRRTLE's stronger performance in some contexts and weaker performance in others."

The above reworded sentence may help eliminate any ambiguity suggesting that HRRTLE's performance is being compared to outputs from other model(s).

RC: Line 491: What was the degree of development in the catchments? Comparing how much "disturbance" is in a catchment could lend insight into the varying degrees of runoff efficiency and therefore how important transmission losses may be in a

catchment.

AC: Thank you for this question and comment. The authors suspect that the extent of "disturbance" could play a significant role in model performance. In a revised version of the manuscript, we would be willing to place further emphasis on the level of disturbance in each catchment. Some thought is required on how to (objectively) quantify the degree of disturbance in each catchment. Extracting urban areas from land cover/land use maps could prove useful. Additionally, quantifying the extent of standing water within each catchment may provide valuable insights.

The additional catchment characteristics mentioned in the above paragraph could also be included in a Principal Components Analysis alongside the catchment characteristics provided in Figure 5 of the original 'pre-print' manuscript.

RC: Lines 535-545: This paragraph seems disorganized and a bit tough to read. This seems like it would be better as a table or reduced to a single line that states "Studies that utilize varying types of hydrologic models (rainfall-runoff, hydrodynamic, process-based, etc) do not explicitly represent transmission losses (citations)." Then transition to why this is important tied to the results of this study. Right now, this reads as a "bashing" of other studies.

AC: The authors acknowledge that this paragraph can and should be improved. We believe it should be reworded to position HRRTLE as building upon the work of others by accounting for transmission losses, while still incorporating key aspects of the aforementioned studies.